

# Accuracy assessment of global internal tide models using satellite altimetry

Loren Carrere[1], Brian K. Arbic[2], Brian Dushaw[3], Gary Egbert[4], Svetlana Erofeeva[4], Florent Lyard[5], Richard D. Ray[6], Clément Ubelmann[7], Edward Zaron[8], Zhongxiang Zhao[9], Jay F. Shriver[10], Maarten Cornelis Buijsman[11], Nicolas Picot[12]

[1]CLS, Ramonville-Saint-Agne, 31450, France
[2]University of Michigan, Ann Arbor, MI, USA
[3]no affiliation, Girona, 17004, Spain
[4]Department Geology and Geophysics, Oregon State University, Corvallis, OR 97331-5503, USA
[5]LEGOS-OMP laboratory, Toulouse, 31401, France
[6]NASA Goddard Space Flight Center, Greenbelt, MD 20771, USA
[7]Ocean Next, La Terrasse, 38660, France
[8]Department of Civil and Environmental Engineering, Portland State University, Portland, OR 97207-0751, USA
[9]Applied Physics Laboratory, University of Washington, Seattle, WA, USA
[10]Naval Research Laboratory, Stennis Space Center, MS, USA
[11]Division of Marine Science, University of Southern Mississippi, Stennis Space Center, MS 39529, USA
[12]CNES, Toulouse, 31400, France

*Correspondence to*: L. Carrere (lcarrere@groupcls.com)

**Abstract.** In order to access the targeted ocean signal, altimeter measurements are corrected for several geophysical parameters among which the ocean tide correction is one of the most critical, but the internal tide signature at the surface are not yet corrected globally.

Internal tides can have a signature of several cm at the surface with wavelengths about 50-250 km for the first mode and even smaller scales for higher order modes. The goals of the upcoming Surface Water Ocean Topography (SWOT) mission and other high-resolution ocean measurements make the correction of these small scale signals a challenge, as the separation of all tidal variability from other oceanic signals becomes mandatory.

In this context, several scientific teams are working on the development of new internal tide models, taking advantage of the very long altimeter time series now available, which represent an unprecedented and valuable global ocean database. The internal tide models presented here focus on the coherent internal tide signal and they are of three types: empirical models based upon analysis of existing altimeter missions, an assimilative model, and a three-dimensional hydrodynamic model.

A detailed comparison and validation of these internal tide models is proposed using existing satellite altimeter databases. The analysis focuses on the four main tidal constituents M2, K1, O1 and S2. The validation process is based on a statistical analysis of multi-mission altimetry including Jason-2 and Cryosphere Satellite-2 data, taking advantage of the long-term altimeter databases available. The results show a significant altimeter variance reduction when using internal tide corrections on all ocean regions where internal tides are generating/propagating. A complementary spectral analysis also gives some estimation



of the performance of each model as a function of wavelength, and some insight into the residual non-stationary part of internal tides in the different regions of interest.

## 1 Introduction

Since the early 1990s, several altimeter missions have been monitoring sea level at a global scale, offering nowadays a long
and very accurate time-series of measurements. This altimetry database is nearly homogeneous over the entire ocean, allowing sampling many regions that were poorly or even not sampled before the satellite era. Thanks to its current accuracy and maturity, altimetry is now considered as a fully operational observing system dedicated to ocean and climate applications (Escudier et al. 2017).

The main difficulty encountered when using altimeter datasets for ocean studies is related to the long revisit time of the
satellites which results in the aliasing of high-frequency ocean signals into a much lower frequency band. Concerning tidal frequencies, the 9.9156-days cycle of Topex-Poseidon/Jason altimeter series induces the aliasing of the semidiurnal M2 lunar tide into a 62 day period, and the diurnal K1 tide is aliased into a 173 days period, the latter of which is very close to the semi-annual frequency and raises complex separation problems. The long duration of the global ocean altimeter database available has allowed the community to overcome this separation problem, and new global ocean barotropic tidal solutions (Stammer at
al. 2014) have been produced taking advantage of altimeter data: among them the last Goddard/Grenoble Ocean Tide model (noted GOT: Ray, 2013) and the last Finite Elements Solution for ocean tide (noted FES2014: Carrere et al. 2016; Lyard et al. 2020) are commonly used as reference for the barotropic tide correction in actual altimeter Geophysical Data Records (noted GDRs). Moreover this altimeter database has been used in numerous studies to validate new instrumental and geophysical corrections used in altimetry, thanks to the analysis of their impact on the sea level estimation at climate scales, as well as at
lower temporal scales like mesoscale signals; particularly it has proven its efficiency for validating global ocean models (Stammer et al. 2014; Carrere et al. 2016; Quartly et al. 2017).

The coming Surface Water Ocean Topography (SWOT) mission, led by NASA, CNES, and the UK and Canadian space agencies, is planned for 2021 and will measure sea surface height with a spatial resolution never proposed before, thus raising the importance of the correction of the internal tide surface signature. Internal tides (denoted IT) are generated by an incoming
barotropic tidal flow on a bathymetric pattern within a stratified ocean, and can have amplitudes of several tens of meters at the thermocline level and a signature of several centimeters at the surface, with wavelengths ranging approximately between 30 and 250 km for the lowest three modes of variability (Chelton et al. 1998). From the perspective of the SWOT mission and of high-resolution ocean measurements in general, removing these small scale surface signals is a challenge, because we need to be able to separate all tidal signals to access other oceanic variability of interest such as mesoscale, sub-mesoscale or climate
signals.

A large part of the internal tide signal remains coherent over long times, with large stable propagation patterns across ocean basins, such as the North Pacific and many other regions. The amplitude of the coherent signal appears to be greatly diminished



in the equatorial regions which may be caused by the direct disrupting effect of the rapid equatorial wave variations (Buijsman

et al. 2017) or merely masked by the background noise. The seasonal variability of the ocean medium and the interaction with mesoscale eddies and currents may also disrupt the coherence of the internal tides in many other areas, which makes the non-coherent internal tides variability more complex to observe and model (Shriver et al. 2014).

In this context, and since conventional satellite altimetry has already shown its ability to detect the small scale internal tide surface signatures (Ray and Mitchum, 1997; Dushaw 2002; Carrere et al. 2004), several scientific teams have developed new internal tides models, taking advantage of the very long altimeter time series now available. These internal tide models are of

three types: empirical models based upon analysis of existing altimeter missions, usually more than one, assimilative models based upon assimilating altimeter- data into a reduced gravity model, and three-dimensional hydrodynamic models, that embed internal tides into an eddying general circulation model. In the present paper, the analysis is focused on seven models that yield a coherent internal tide solution: Dushaw 2015, Egbert and Erofeeva 2014, Ray and Zaron 2016, Shriver et al. 2014, Ubelmann (personal communication, 2017), Zaron 2019, Zhao et al. 2016.

The objective of this paper is to present a detailed comparison and a validation assessment of these internal tide models using satellite altimetry. The analysis focuses on the correction of the satellites' measurements from the coherent internal tide signal for the main tidal constituents, M2, S2, K1 and O1. The validation process is based on a statistical analysis and on a comparison to multi-mission altimetry including Jason-2 (noted J2 hereafter), AltiKa and Cryosphere Satellite-2 data (also named Cryosat-2 or C2 hereafter), taking advantage of the various and long-term altimeter databases available. For the sake of clarity, only

results for Jason-2 and Cryosat-2 altimeters and for the main tidal components M2 and K1 are presented in the core of this paper, and O1 and S2 validation results are gathered in the appendix.

After a brief description of the participating models (section 2), an analysis of the differences between internal tide models is proposed in section 3. Section 4 describes the altimeter dataset used, the method of comparison and the validation strategy. The validation results of the different internal tide corrections versus altimetry databases are described in sections 5 and 6.

Finally, a discussion and concluding remarks are gathered in section 7.

**2. Presentation of participating internal tide models**

This section gives a brief overview of the internal tide models evaluated in this study. We considered five purely empirical models involving data merging, one data assimilative model and also one pure hydrodynamic model simulating tides and internal tides using the gravitational forcing and a high spatial resolution but without any internal tide data constraint. The list

of participating IT models is given in Table 1.

| Model name | Type of model | Constituents tested | Altimeter data used | Authors |
|---|---|---|---|---|
| DUSHAW | E | M2 | TP+J1 | Dushaw, 2015 |



| EGBERT | A | M2, K1, O1, S2 | ERS-EN+TP-J1-J2 | Egbert and Erofeeva, 2014, 2002 |
|---|---|---|---|---|
| HYCOM | H | M2 | - | Shriver et al. 2014 |
| RAY | E | M2 | GFO+ERS-EN+TP-J1-J2 | Ray and Zaron 2016 |
| UBELMANN | E | M2 | All except C2 | Ubelmann et al., in prep. |
| ZARON (HRET) | E | M2, K1, O1, S2 | TP-J1-J2+ERS-EN-AL+GFO | Zaron 2019 |
| ZHAO | E | M2, K1, O1, S2 | GFO+ERS-EN+TP-J1-J2 | Zhao et al. 2016 |

**Table 1 : List of the participating IT models. Most of the models are global models except one that is currently available in only 2 areas (Hawaii and Azores, noted in gray). E = Empirical model; A = Assimilative model; H = Hydrodynamic model. Acronyms used for altimeter missions: TP=Topex/Poseidon; J1 = Jason-1; J2 = Jason-2; EN = Envisat; GFO = Geosat Follow On; C2 = Cryosat-2; AL = AltiKa**

## 2.1 Empirical models

The purely empirical models are based upon the analysis of existing conventional altimeter missions, usually more than one. The five empirical models used in the present study are briefly described below.

- **DUSHAW**

This global model was computed using a frequency-wavenumber tidal analysis (Dushaw et al. 2011). The internal tides were assumed to be composed of narrow-band spectra of traveling waves, and these waves are fit to the altimeter data in both time and position. A tidal analysis of a simple time series can extract accurate tidal estimates from noisy or irregular data of enough long record under the assumptions that the signal is temporally coherent and described by a few known frequencies. The frequency-wavenumber analysis generalizes such an analysis to include the spatial dimension, making the strong assumptions that both time and spatial wave variations are coherent. In addition to the known tidal constituent frequencies, the solution also requires accurate values for the local intrinsic wavelengths of low-mode internal waves. Internal tide properties, which depend on inertial frequency, stratification and depth, were derived using the 2009 World Ocean Atlas (Antonov et al. 2010, Locarnini et al. 2010) and Smith and Sandwell global seafloor topography (Smith and Sandwell, 1997). The solution is a spectral model with no inherent grid resolution; tidal quantities of interest derived from the solution are both inherently consistent with the data employed and influenced by non-local data.

The fit used M2, S2, N2, K2, O1, and K1 constituents, with spectral bands for barotropic, mode-1 and mode-2 wavenumbers. Data from T/P and Jason-1 altimetry programs were employed. These data had the barotropic tides removed, but the fit allowed for residual barotropic variations. Employing all constituents and wavelengths simultaneously in a single fit minimized the chance that the solution for a particular constituent was influenced by noise from nearby tidal constituents. To account for





regional variations of the internal tide characteristics (and reduce computational cost) independent fits were made in 11°×11° overlapping regions. The global solution was obtained by merging the regional solutions together using a cosine taper over a 1° interval; the solution is therefore sometimes discontinuous within these overlapping zones. For this study, global maps of

the harmonic constants for the two first baroclinic modes of the largest semi-diurnal tidal constituent M2 were computed on a regular 1/20° grid (Dushaw, 2015; the complete solution is available from http://www.apl.washington.edu/project/project.php?id=tm_1-15 ). This global M2 solution was tested against pointwise, along-track estimates for the internal tide, with satisfactory comparisons in the Atlantic and Pacific oceans. Comparisons were also made to in situ measurements by ocean acoustic tomography in the Pacific and Atlantic, showing a good predictability in

both amplitude and phase. By comparisons to the tomography data, internal tides within the Philippine Sea or Canary Basin were less predictable. Some of these comparisons found good agreement between hindcasts and time series recorded in the western North Atlantic about a decade before the altimetry data were available, which is consistent with the extraordinary temporal coherence of this IT signal in many regions of the world's oceans.

• **RAY**

RAY model provides a global chart of surface elevations associated with the stationary M2 internal tide signal. This map is empirically constructed from multi-mission satellite altimeter data, including GFO, ERS, ENVISAT, Topex/Poseidon, Jason-1 and Jason-2 missions. Although the present-day altimeter coverage is not entirely adequate to support a direct mapping of very short-wavelength features such as surface internal tides signatures, using an empirical mapping approach produces a

model that is independent of any assumption about ocean wave dynamics. Validation using some independent data from CryoSat-2 showed a positive variance reduction in most areas except in regions of large mesoscale variability, due to some contamination from non-tidal ocean variability in these last regions (Ray and Zaron, 2016). In the model version used here, those regions have been masked with a taper to give zero elevation. The model grid has a 1/20° resolution and it is defined over the 50°S - 60°N latitude band.


• **UBELMANN**

The internal tide solution is obtained from all altimetry satellites in the period 1990-2013, except for the Cryosat-2 mission. The method relies on a simultaneous estimation of the mesoscales and coherent M2 internal tides. Indeed, the mesoscale signal is known to introduce errors in the tidal estimation (non-zero harmonics on a finite time window). To mitigate that issue, most

existing methods subtract the low-frequency altimetry field from AVISO as a proxy for mesoscales (e.g. Ray and Zaron 2016). However, the estimate of the mesoscale is itself contaminated by internal tides (e.g. Zaron and Ray, 2018) aliased into low frequency which also introduces errors. For these reasons, Ubelmann proposed here a simultaneous estimation, accounting for the covariances of mesoscales and internal tides in a single inversion. In practice, these covariances are expressed in a reduced wavelet basis (local in time and space) for mesoscales and in a plane wave basis (local in space only) for internal tides. The



plane wave wavelength and phase speed rely on the 1st and 2nd Rossby radii of deformation climatology by Chelton et al., 2001. Although the inversion cannot be done explicitly (because of the long time-window extending the basis size for mesoscale), a variational minimization allows for a converged solution after about 100 iterations (typical degree of freedom for the problem). For this study, only the M2 internal tide solution (for mode 1 and mode 2) is considered, but the mesoscale solution is also of interest because the internal tide contamination should be minimized compared to the standard AVISO

processing.

The method is being further described in Ubelmann et al., in prep. Further improvements are expected after introducing additional tide components in the same inversion, and after considering slow (or seasonal) variation of the phases.

- **ZARON**

The High Resolution Empirical Tide (HRET) model provides an empirical estimate for the baroclinic tides at the M2, S2, K1, O1 frequencies, as well as the annual modulations of M2, denoted MA2 and MB2. The development of HRET begins with assembling time series of essentially all the exact-repeat mission altimetry along the reference and interleaved orbit ground tracks of the TOPEX/Poseidon--Jason missions, the ERS--Envisat--AltiKa missions, and the Geosat Follow-On mission (Zaron, 2019). Standard atmospheric path delay and environmental corrections are applied to the data, including removal of

the barotropic tide using the GOT4.10c model and removal of an estimate for the mesoscale sea level anomaly using a purpose-filtered version of the Ssalto/Duacs multi-mission L4 sea level anomaly product (Zaron and Ray, 2018). Conventional harmonic analysis is then used to compute harmonic constants at each point along the nominal 1-Hz ground tracks (Carrere et al., 2014), and these data are used as inputs for subsequent steps.

HRET was initially developed to evaluate plausible spatial models for the baroclinic tides, seeking ways to improve on some

previous models (Zhao et al., 2012; Ray and Zaron, 2016). It uses a local representation of the wave field as a sum of waves modulated by an amplitude envelope consisting of a second-order polynomial, thus generalizing the spatial signal model used in previous plane-wave fitting (Ray and Mitchum, 1996; Zhao et al, 2016). The details of the implementation in HRET differ in additional ways from previous approaches. Specifically, the wavenumber modulus and direction of each wave component are determined by local 2-dimensional Fourier analysis of the along-track data, and the coefficients in the spatial model are

determined by weighted least-squares fitting to along-track slope data--the latter removes the need for rather arbitrary along-track high pass filtering used in other estimates. Hence, the model is fully empirical in the sense that it does not use an a priori wavenumber dispersion relation.

The above-described approach to building local models for the baroclinic waves is applied to overlapping patches of the ocean, which are then blended and smoothly interpolated on a uniform latitude-longitude grid. Using the standard error estimates

from the original harmonic analysis and goodness-of-fit information from the spatial models, a mask is prepared which smoothly damps the model fields to zero in regions where the estimate is believed to be too noisy to be useful. These are generally regions near the coastline where the number of data used are reduced, or regions in western boundary currents or the





Southern Ocean where the baroclinic tides cannot be distinguished from the continuum of energetic mesoscale variability. HRET version 7.0 was provided for the present validation analysis. Note that the model is still being refined and version 8.1

is available at present: it has improved O1 relative to the results shown here, and made minor changes to the other constituents.

- **ZHAO**

This model is constructed by a two-dimensional plane wave fit method (Zhao et al. 2016). In this method, internal tidal waves are extracted by fitting plane waves using SSH measurements in individual fitting windows (160 km by 160 km for M2).

Prerequisite wavenumbers are calculated using climatological ocean stratification profiles. For each window, the amplitude and phase of one plane wave in each compass direction (angular increment 1°) are determined by the least-squares fit. When the fitted amplitudes are plotted as a function of direction in polar coordinates, an internal tidal wave appears to be a lobe. The largest lobe gives the amplitude and direction of one internal tidal wave. The signal of the determined wave is predicted and removed from the initial SSH measurements. This procedure can be repeated to extract an arbitrary number of waves (3 waves

here). Four tidal constituents M2, S2, O1 and K1 are mapped separately using their respective parameters and are used in the present paper (model version Zhao16). This mapping technique dynamically interpolates internal tidal waves at off-track sites using neighboring on-track measurements, overcoming the difficulty posed by widely-spaced ground tracks. There are a large number of independent SSH measurements in each fitting window, compared to a single time series of SSH measurements used by point-wise harmonic analysis. As a result, nontidal noise caused by tidal aliasing can be significantly suppressed. This

technique resolves multiple waves of different propagation directions; therefore, the decomposed internal tide fields may provide more information on generation and propagation.

### 2.2 Assimilative model

G. Egbert and S. Erofeeva have developed a reduced gravity (RG) data assimilation scheme for mapping low-mode coherent internal tides (Egbert and Erofeeva, 2014), and applied this to a multi-mission dataset to produce global first mode M2 and K1

solutions. This scheme is based on the Boussinesq linear equations for flow over arbitrary topography with a free surface and horizontally uniform stratification. As in Tailleux and McWilliams (2001) and Griffiths and Grimshaw (2007), vertical dependence of the flow variables are described using flat-bottom modes (which depend on the local depth H(x, y)), yielding a coupled system of (2-dimensional) PDEs for the modal coefficients for surface elevation and horizontal velocity. Equations for each mode are coupled through interaction coefficients, which can be given in terms of the vertical mode eigenvalues

following the approach of (Griffiths and Grimshaw, 2007). Modes are decoupled wherever bathymetric gradients are zero, and for a flat bottom the system reduces to the usual single mode RG shallow water equations.

Within the RG scheme used, the vertical-mode coupling terms are dropped to obtain independent equations for the propagation of each mode with spatially variable reduced water depth, which are determined from local bathymetry and stratification. These simplified equations are identical to the linear shallow water equations used in OSU Tidal Inversion Software (OTIS,



https://www.tpxo.net/otis ; Egbert and Erofeeva, 2002) , thus allowing use of the assimilation system to map internal tides by simply modifying depth, and fitting along-track harmonic constants as a sum over a small number of modes. With some extensions to OTIS, coupling terms for the first few modes can be included in the dynamics.

This OTIS-RG assimilation scheme has been applied to construct global maps of first mode temporally coherent internal tide elevations. Available exact repeat mission data, except Geosat-Follow-On (GFO), were assimilated (TP/Jason, ERS/Envisat),

with the AVISO weekly gridded SSH product used to reduce mesoscale variations before harmonic analysis. Solutions are computed in overlapping patches (~ 20 x 30 degrees), and then merged (via weighted average on overlaps) into a global solution. It can be noticed that adjacent solutions almost always match quite well even without this explicit tapering.

## 2.3 Hydrodynamic model

The hydrodynamic internal-tide solution is provided by the three-dimensional ocean model HYCOM (HYbrid Coordinate

Ocean Model), that embed tides and internal tides into an eddying general circulation model (Shriver et al. 2014). A free simulation, i.e. without any data assimilation (simulation n°102), is used for the present study; this run used an Augmented State Ensemble Kalman Filter (ASEnKF) to correct the forcing and reduce the M2 barotropic tidal error to about 2.6 cm (Ngodock et al., 2016). The value of such a simulation is to provide some information about the interaction of internal tides with mesoscales and other oceanic signals in addition to the internal tides signal itself, which means that it can give access to

the non-coherent internal tide signal too. For the present study, a one-year simulation has been run and a harmonic analysis of the steric SSH allowed extraction of the M2 internal tide signal which remains coherent on this period. The non-assimilative quality of the simulation makes it entirely independent from the altimeter database used for the validation. The spatial resolution of the native grid is 1/24°.

## 3. Comparison of internal tide (IT) models

3.1 Qualitative comparison of IT elevations

A first analysis of the model differences consists in visualizing the patterns of IT models' amplitude on the regions of interest defined on Figure 1. These seven regions are characterized by a well-known and nearly permanent internal tide signal, already pointed out by previous studies (Egbert et al. 2000, Carrere et al. 2004, Nugroho 2017).





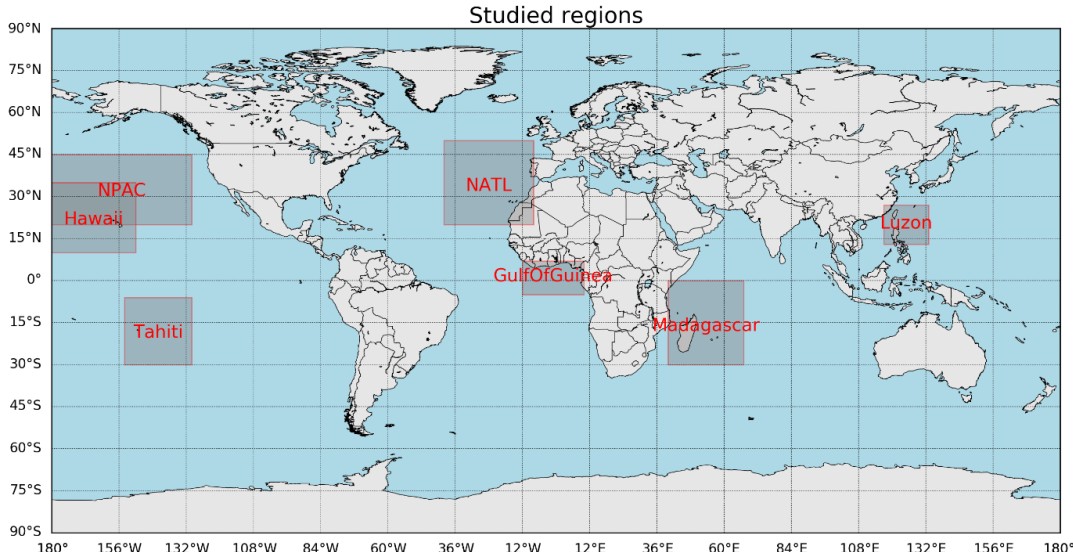

**Figure 1 : Localization of the internal tide regions studied in the present paper.**

Figure 2 shows the M2 IT amplitude of each model in the North Pacific area (NPAC) located around the Hawaiian Islands. In this region, all models have similar amplitudes and similar beam patterns demonstrating north-eastward propagation with one clear northward beam; amplitudes are often greater than 2 cm. The amplitude's pattern varies along IT beams with short spatial scales, indicating that most of the models capture a part of the higher order IT modes: typical 70 km patterns are visible corresponding to the 2nd M2 IT mode wavelength in this region. The ZHAO solution shows cleaner and smoother patterns likely due to the theoretical plane wave approximation used for the estimation. RAY, ZHAO, and EGBERT propagate until 150°W while ZARON propagates farther to the east and EGBERT has the most attenuated amplitudes on the region. UBELMANN and DUSHAW models show similar patterns but both maps are noisier compared to other solutions. HYCOM also shows similar beams but with clearly stronger amplitudes, and some noise is also noticeable on the maps.

M2 IT amplitudes in the Luzon region are plotted on Figure 3. Only 6 models are plotted as UBELMANN is not defined on this area. The models have an M2 amplitude greater than 2 cm in the Luzon region, and HYCOM is significantly stronger than the other models. The IT propagation pattern also shows small spatial scales (of the order of 100 km eastward of the strait) indicating that higher IT modes are also enhanced at the semi-diurnal frequency, but the models do not agree on a clear common pattern: DUSHAW has a rather noisy structure and a discontinuity appears along longitude 125°E due to the effect of the different computational patches used to estimate the global solution. All other models show a strong M2 amplitude across the Luzon strait; on the east side of the strait, two beams respectively northward and southward along the Taiwan and the Philippines islands are visible, and a wide eastward beam is visible in the ZARON, ZHAO and HYCOM maps. The patterns are noisier for the EGBERT and RAY solutions. The ZARON and HYCOM solutions are close to zero in shallow waters, while RAY, ZHAO and EGBERT are not defined; DUSHAW is defined in shallow waters showing some propagation patterns,





but one must be careful as an empirical model might have difficulties to separate IT surface signatures from small scales of barotropic tides occurring in these areas. At the strait itself the main wave propagation is expected to be predominantly in the west and/or east directions, which is challenging for empirical techniques to recover owing to the primarily north-south

270     altimeter track orientations. The problem was discussed in some detail by Ray and Zaron (2016), and indeed their model has very little eastward-propagating energy from the strait (see also Zhao, 2019). Plots of the M2 IT for other regions defined in Figure 1 are provided as supplementary material.







Figure 2 : amplitude of the IT models for M2 tide component on NPAC region (north Hawaii)





275

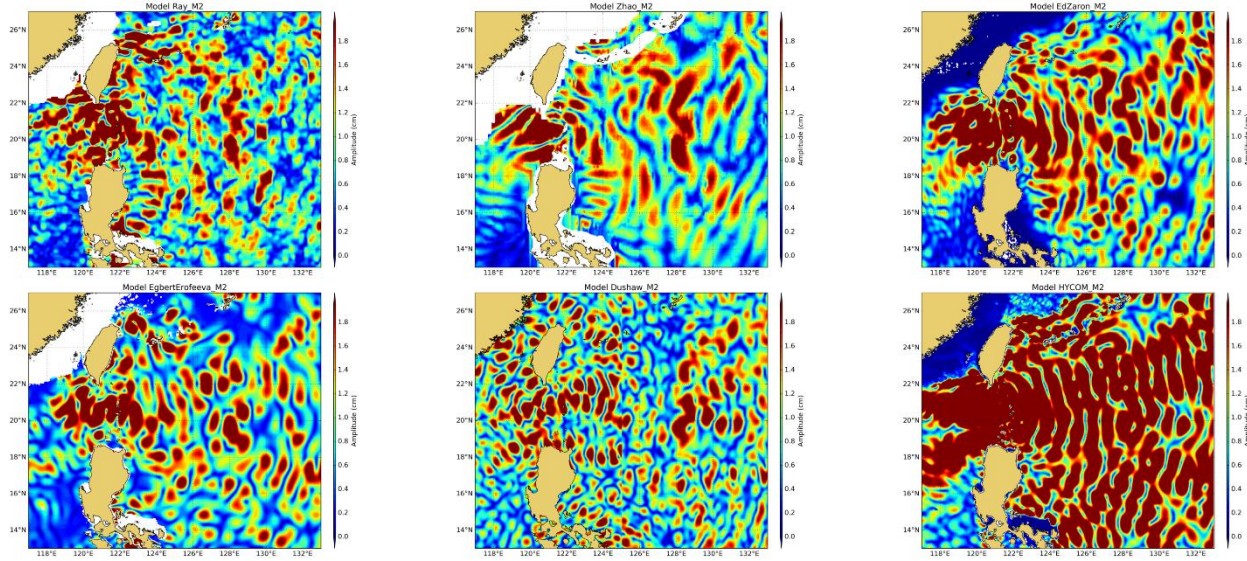

**Figure 3 : amplitude of the IT models for M2 tide component on Luzon area**

Figure 4 shows the amplitude of the 3 IT solutions available for K1 wave on the Luzon region, where amplitudes of the diurnal

280 IT are the most important. Models show large scale (about 200 km or more) patterns on both sides of the Luzon strait. The

K1 scales are significantly greater than M2 scales as expected from theoretical wavelengths. The K1 amplitude reaches 2 cm

on the west side, while patterns and amplitudes of the models differ on the east side of the strait: ZHAO has weaker amplitudes

and some different spatial patterns, while ZARON and EGBERT have the solutions that lie closest one to each other. For these

3 models, the amplitude of K1 becomes zero at about 24°N when getting close to the K1 critical latitude.

Concerning diurnal tides in the global ocean, the ZARON solution is not defined over large regions of the world ocean,

including latitudes poleward of the diurnal tide critical latitude and regions where the IT amplitude is negligible and/or not

separable from background ocean variability. The ZHAO solution stops at the diurnal critical latitude, while the EGBERT

solution is defined over a wider range of latitudes (until 60°).

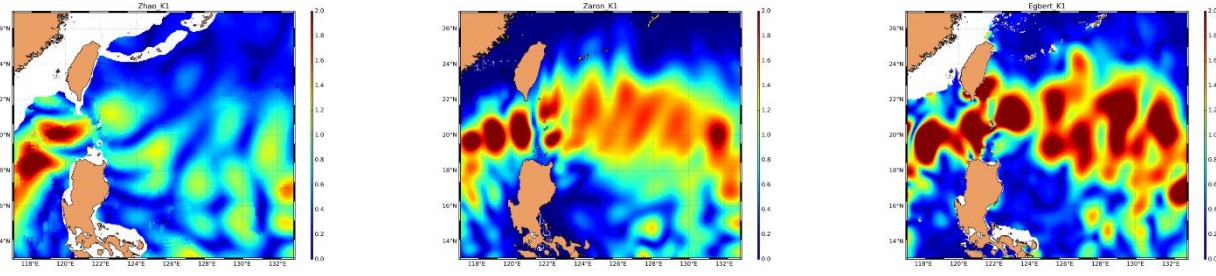


**Figure 4 : amplitude of the IT models for K1 tide component, on the Luzon area.**





### 3.2 Quantitative comparison of IT models

Following Stammer et al. (2014), the standard deviation (STD) of the IT elevations in the models listed in Table 1 can be
computed. The computation of the STD was performed for the four tidal constituents M2, S2, K1 and O1, after re-gridding
bilinearly the models to a common 1/20° grid. The maps of STD are computed over the global ocean. Notice that the DUSHAW
model was not included in this STD calculation, as it increases too much the STD value over the global ocean and makes the
results difficult to analyze.

Global maps shown in Figures 5 and 6 illustrate respectively the mean and the standard deviation of the M2 and K1 IT models.
Near-coastal regions, shallow water regions, and regions of low signal-to-noise are masked-out on the maps as they are not
defined in most of the studied models. The mean M2 amplitudes reach more than 2 cm in all the known generation sites--in
the Pacific, the Indian Ocean around Madagascar, the Indonesian Seas, and in the Atlantic offshore of Amazonia.  K1 has a
significant mean amplitude above 1.5 cm in the Luzon strait region, in the Philippine Sea and east of Palau, and about 0.5-0.7
cm in some regions of the Indian and Pacific oceans.

The map of M2 standard deviation shows small values, generally below 1 cm for M2, indicating a good agreement of the IT
models in all IT regions defined in Figure 1 for the M2 wave; some larger standard deviation values are found around Luzon
strait, above Madagascar and in the Indonesian seas. For the diurnal wave K1, IT models provide coherent information in the
Luzon region, in Tahiti and Hawaii and also in the Madagascar region.

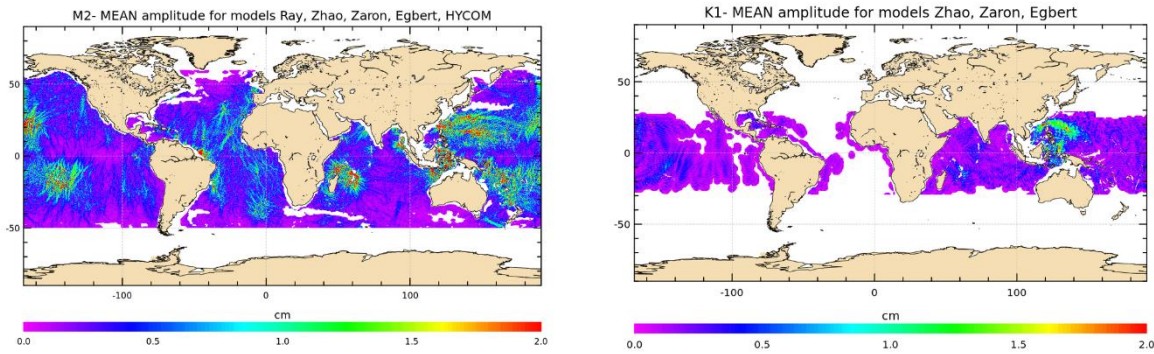

**Figure 5 : global maps of mean amplitude of the M2 and K1 IT models (cm)**




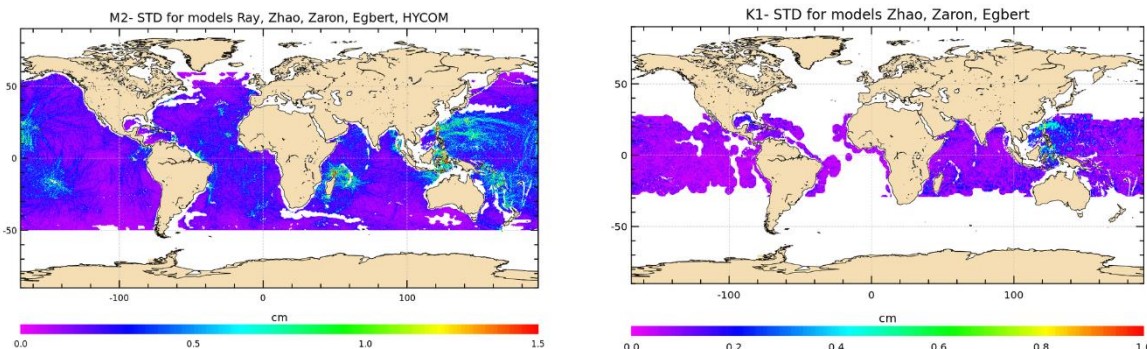

**Figure 6 : global maps of standard deviation of the M2 and K1 IT models (cm)**

The mean standard deviation value is computed over the different regions studied. In order to eliminate any residual barotropic

variability likely existing in the empirical IT models in shallow waters, only data located in deep ocean are used to compute

the standard deviation; values are gathered in Table 2. Over all regions, the standard deviation is stronger for M2, consistent

with the fact that M2 is the most important IT component on the global ocean. The standard deviation is largest in the Luzon

and Madagascar regions, where models give rather different solutions as already seen in the previous section.

The diurnal K1 tide takes on the largest standard deviation value, of 0.25 cm, in the Luzon region, where this diurnal component

has the most significant amplitudes.

| STD for deep ocean (cm) | M2 | K1 |
|---|---|---|
| Tahiti | 0,36 | 0.07 |
| Hawaii | 0,33 | 0.07 |
| Madagascar | 0,46 | 0.10 |
| Gulf of Guinea | 0,21 | 0.07 |
| Luzon | 0,54 | 0.25 |
| NATL | 0,15 | - |
| NPAC | 0,20 | - |

**Table 2 : Spatial-mean standard deviation (cm) of the M2 and K1 IT models for each studied region.**



## 4. Presentation of the altimeter database and the method of comparison

### 4.1 The altimeter database

The altimeter measurements used correspond to the level-2 altimeter products L2P, with 1-Hz along-track resolution, produced and distributed by Aviso+ (https://www.aviso.altimetry.fr/en/data/products/sea-surface-height-products/global/along-track-sea-level-anomalies-l2p.html, AVISO), as part of the Ssalto ground processing segment. The version of the products considered is nearly homogeneous with the DT-2014 standards described in Pujol et al (2016), except for the tide correction as described below.

The altimeter period from 1993 onwards is sampled by twelve altimeter missions available on different ground tracks (https://www.aviso.altimetry.fr/en/missions.html). For the purpose of the present study, we use the databases for two different missions:

- Jason-2 (noted J2 in the text and figures) is a reference mission flying on the reference TP track with a 10-day cycle and sampling latitudes between +/-66°; the entire mission time-span on the reference track can be used for the study
which represents nearly 8 years of data;
- Cryosat-2 (noted C2 hereafter) is characterized by a drifting polar orbit sampling all polar seas and it has a nearly repetitive sub-cycle of about 29 days.

The mission's time series and the number of cycles used for the present study are listed in Table 3. It is worth pointing out that much of the T/P and Jason- data have been used in most of the IT empirical solutions tested (cf Table 1), but all models are 340 independent of Cryosat-2 mission data.

Due to a sub-optimal time sampling, altimeters alias the tidal signal to much longer periods than the actual tidal period. The aliased frequencies of the 4 main tidal waves studied are listed in Table 3 for the 2 orbits used. It is noticeable that the diurnal tide K1 is the trickiest to observe with satellite altimetry as it is aliased to the semi-annual period by Jason- orbit and to a nearly 4 years period by the C2 satellite orbits. C2 aliasing periods are very long compared to Jason's ones.


| Mission | J2 | C2 |
|---|---|---|
| Repeat period  (days) | 9.9156 | sub-cycle of 28.941 |
| Cycles used | 1-288 (8 years) | 14-77 (5 years) |
| Darwin name | Aliasing (days) | Aliasing (days) |
| $O_1$ | 45.7 | 294.4 |
| $K_1^L$ | 173.2 | 1430 |
| $M_2$ | 62.1 | 370.7 |
| $S_2$ | 58.7 | 245.2 |

**Table 3 : Description of the altimeter database for the validation study, along with the associated aliasing periods for the main tidal components.**





The altimeter sea surface height (SSH) is defined as the difference between orbit and range, corrected from several instrumental
and geophysical corrections as expressed below:

$$SSH = orbit - range - Tide - \textbf{IT} - Other\_corr$$

where

- Tide includes the geocentric barotropic tide, the solid Earth tide, and the pole tide corrections. The geocentric
barotropic tide correction was updated compared to the altimetry standards listed in Pujol et al. (2016), and
355         comes from the FES2014b tidal model (https://www.aviso.altimetry.fr/en/data/products/auxiliary-
products/global-tide-fes/description-fes2014.html ; Carrere et al 2016; Lyard et al. in preparation);
- **IT** is the internal tide correction, taken one-by-one from each model studied in this paper;
- Other_corr includes the Dynamic Atmospheric Correction, the Wet Tropospheric Correction, the Dry
Tropospheric Correction, the Ionospheric Correction, the Sea State Bias Correction, and complementary
360         instrumental corrections when needed, as described in Pujol et al. (2016).

The sea level anomaly (SLA) is defined by the difference between the SSH and a mean profile (MP) for repetitive orbits or a
mean sea surface (MSS) for drifting orbits. Mean profiles computed for Topex/Jason orbit for the reference period of 20 years
(1993–2012), have been used within the present study for Jason-2 mission (Pujol et al. 2016), and the MSS_CNES_CLS_11
also   referenced   on   the   same   20   years   period   was   used   for   the   C2   drifting   orbit   mission
(https://www.aviso.altimetry.fr/en/data/products/auxiliary-products/mss.html ; Schaeffer et al. 2012, Pujol et al. 2016-
Appendix-A).

**4.2 Method of comparison**

Satellite altimetry databases can be used to evaluate many geophysical corrections and particularly global barotropic tidal
models as already examined by other authors (Stammer et al. 2014, Carrere et al. 2012, Lyard et al. 2006, Carrere 2003). We
propose to use a similar approach to validate the concurrent IT models listed in Table 1.

The first step consists in generating the corresponding IT correction for each along-track altimeter measurement, computed
from the interpolation of each IT atlas onto the satellites' ground tracks and the use of a tidal prediction algorithm. Each tidal
component is considered separately for the clarity of the analysis, keeping in mind that the various IT models do not all contain
the same waves.

The altimeter SSH using successively each of the IT corrections tested can then be computed, and the differences in the sea
level contents are analyzed for different time and spatial scales. In particular, considering several altimeters allows the study
of different temporal periods.  As the missions considered, J2 and C2, have different ground tracks and different orbit (cycle)
characteristics, several aliasing characteristics are tested.

The impact of each IT model on SSH can first be estimated for short temporal scales (time lags lower than 10 days), which are
the main concern here as we consider the main high-frequency tidal components M2, K1, O1, S2. Moreover, these short



temporal scales impact also climate studies since high temporal frequency errors increase the formal estimation error of long-time-scale signals (Ablain et al., 2016; Carrere et al, 2016).

The impact of using each of the studied corrections on the SSH performances is estimated by computing the SSH differences between ascending and descending tracks at crossovers of each altimeter, successively using the studied correction and the reference ZERO correction. Crossover points with time lags shorter than 10 days within one cycle are selected in order to minimize the contribution of the ocean variability at each crossover location. As coherent internal tides have short temporal autocorrelation scales, this diagnostic permits a good estimation of the efficiency of the IT models to reduce the high-frequency variability of the altimeter SSH, focusing on SSH signals with periods below 10 days in the case of this crossovers' diagnostics.

The maps of the variance of SSH differences at crossover points are computed on boxes of 4°x4° holding all measurements within the time span of the mission considered: they give information on the spatio-temporal variance of the SSH differences within the boxes. As SSH differences are considered, this variance estimation is twice the variance difference of SLA. A reduction of this diagnostic indicates an internal consistency of sea level between ascending and descending passes within a 10-day window and thus characterizes a more accurate estimate of SSH for high-frequencies. However, the spatial resolution of this diagnostic is limited due to the localization of crossovers and the 4° resolution of the grid. Particularly for C2, the mission ground-tracks' pattern induces a non-homogeneous spread of crossovers over the global ocean, with no crossovers around latitudes 0° and +/-50°. For J2, all altitudes are covered with crossovers but the number of points is limited at the equator and increases towards the poles.

Along-track SLA statistics can be calculated from 1 Hz altimetric measurements and allow for a higher spatial resolution in the analysis. The maps of the variance difference of SLA using the IT correction tested and the reference ZERO correction are computed on boxes of 2°x2°. Although high-frequency signals are aliased in the lower-frequency band following the application of the Nyquist theory to each altimeter sampling, SLA time series contain the entire ocean variability spectrum. The SLA variance reduction diagnostic shows an improvement of the studied IT correction, on the condition that the correction is decorrelated from the sea level.

The mean of these variance reduction estimations at crossovers and for along-track SLA is computed for each studied region, which allows an easier analysis and comparison of the performances of the IT model tested.

Finally, in order to quantify the impact of each IT model on the SLA variance reduction in terms of spatial scales, a spectral analysis of J2 SLA is performed on the different regions of interest, and details are given in section 6.

## 5. Variance reduction analysis using satellite altimeter data

This section gathers the validation results of each IT model using the satellite altimetry databases described previously. For the clarity of the analysis, each IT correction is compared to a reference correction using a ZERO correction. For the ZERO correction, no IT correction is applied, as in the actual altimeter GDR-D and GDR-E processing (Pujol et al. 2016; Taburet et





al. 2019). The complete diagnostics and analysis are presented hereafter for the largest semidiurnal (M2) and diurnal (K1) components; results for the second largest semidiurnal (S2) and diurnal (O1) IT are gathered in the appendix of the paper.

## 5.1 M2 component


To investigate and quantify the regional impact of the M2 IT corrections, the maps of SSH variance difference at crossovers successively using each IT correction and a ZERO reference correction, are plotted for the J2 mission in Figure 7. Most of the IT models reduce the altimeter SSH variance in all IT regions. The RAY and ZARON models are the most efficient, with a variance reduction reaching more than 5 cm² in many areas. The HYCOM and DUSHAW models reduce SSH variance in

some locations but also raise the variance in others. Mean values, averaged over the strong IT regions shown in Figure 1, are listed in Table 4: the more energetic areas for M2 IT seem to be the Luzon strait and the Hawaii regions with a mean SSH variance reduction greater than 2 cm² for the ZARON model. The ZARON model is the most efficient in all areas except in the NATL region where the UBELMANN model reduces slightly more variance. Over the global ocean, the EGBERT, ZARON, ZHAO and RAY models have similar mean performances, but RAY reduces a bit more the J2 variance globally

(0.34 cm²).

Figure 8 displays the maps of along-track J2 SLA variance differences using successively each M2 IT correction and a ZERO reference correction. Spatial patterns are similar to those in Figure 7. However, using the along-track SLA allows for a better spatial resolution in the output variance maps. In addition, regions of strong IT and regions of strong ocean currents are more

clearly identified. The DUSHAW model raises SLA variance in several mesoscale regions (Gulf Stream, Agulhas current, Malvinas region and Kuroshio currents); ZHAO model also raises slightly the variance in those regions while EGBERT reduces the SLA variance in Gulf Stream and Agulhas regions. HYCOM raises the variance over wider regions in the three oceans than the empirical and assimilative models do. These maps also indicate that the four models, RAY, EGBERT, ZARON and ZHAO, reduce significantly the SLA variance in some additional IT areas which are not specifically investigated in the

present study: the Indonesia seas and south of Java island, north of Sumatra, between Salomon islands and New Zealand in Pacific, off the Amazonian shelf and in many regions of the Atlantic ocean. Mean values, averaged over the strong IT regions identified in Figure 1, are given in Table 4: mean J2 SLA variance reductions are weaker than the crossover differences variances by construction, but they indicate similar conclusions as for J2 crossovers differences: the ZARON model is the most efficient to reduce the SLA variance in all IT regions, except in NPAC and NATL where the UBELMANN model is

slightly more efficient. Mean values over the global ocean are close for the four models EGBERT, ZHAO, ZARON and RAY, with the two last ones showing a slightly better performance than others.





**Figure 7 : Maps of SSH variance differences at crossovers successively using each M2 IT correction and a ZERO reference correction in the SSH calculation for J2 mission (cm2).**






**Figure 8 : Maps of SLA variance differences successively using each M2 IT correction and a ZERO reference correction in the SLA calculation for J2 mission (cm2).**





One should notice that those Jason-2 results might be biased in favor of the empirical models, as Jason-2 data are used in all of them except for the DUSHAW model (cf. Table 1). To check these results, similar diagnostics are computed using the C2 altimeter database, as described in section 4.1, which is an independent database for all models. Validation results are given in Figures 9 and 10 for C2 SSH crossovers differences and C2 SLA respectively.

Validations with the C2 database show similar results as for J2, with a significant variance reduction of the C2 SSH differences and SLA for most models in all IT regions; variance gain patterns are generally similar but wider spread and stronger in C2 SSH maps compared to J2 particularly in the Atlantic ocean and in the west Pacific. The pattern is different for the UBELMANN model in the NATL region, likely due to some inclusion of J2 errors/signal or larger scales signals in the model (cf. section 6). The ground-track pattern of the C2 orbit explains the lack of crossover data at 0° and +/-50° latitudes bands.

C2 SLA variance maps have similar patterns compared to J2, and some additional IT regions are pointed out, which corroborates the quality of the different IT models tested. Over both C2 SSH and SLA, the HYCOM and DUSHAW models show a significant addition of variance in some regions, similarly as for J2 results.

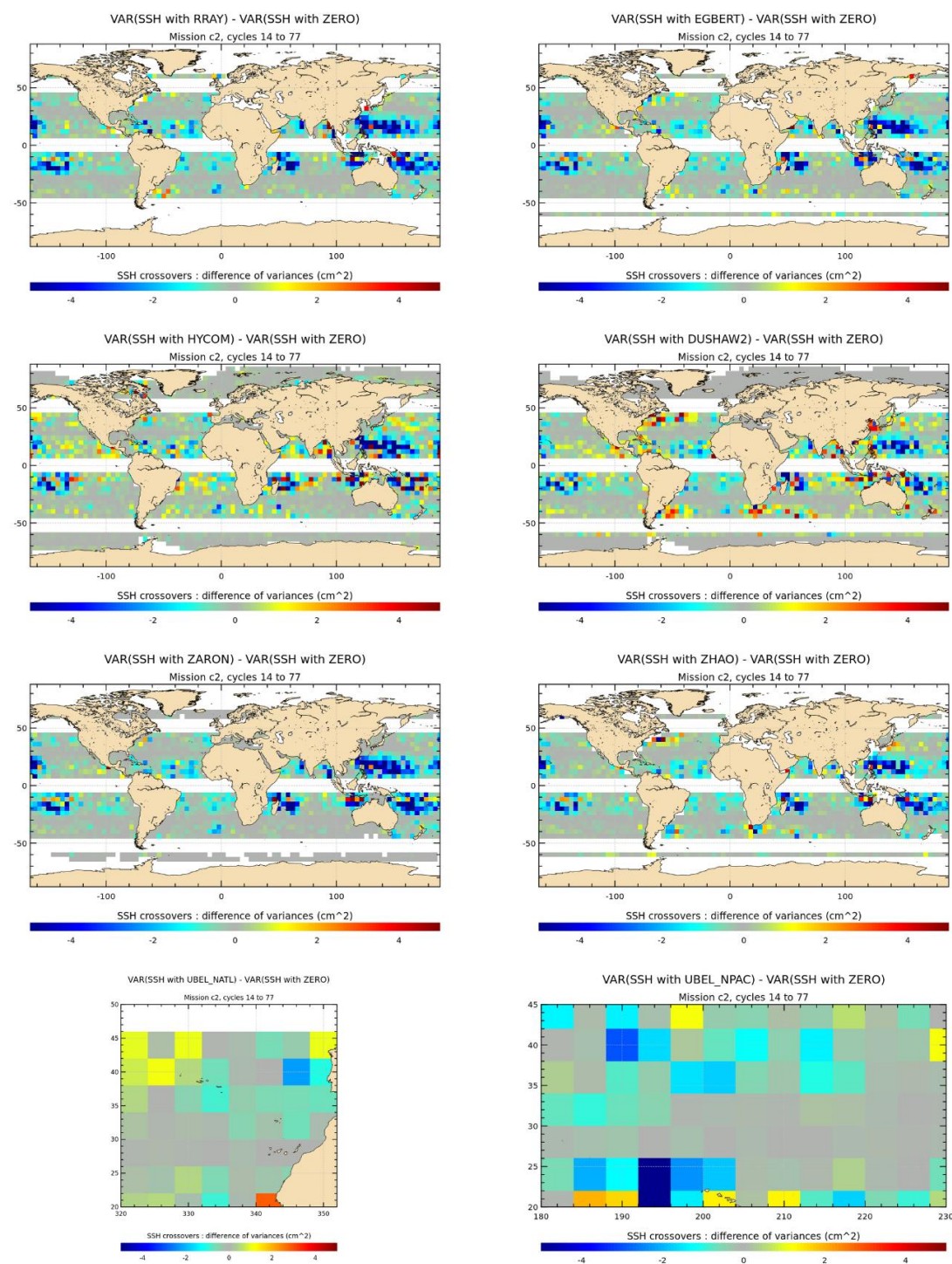

**Figure 9 : Maps of SSH variance differences at crossovers successively using each M2 IT correction and a ZERO reference correction in the SSH calculation for C2 mission (cm2).**





**Figure 10 : Maps of SLA variance differences successively using each M2 IT correction and a ZERO reference correction in the SLA calculation for C2 mission (cm2).**





Mean values for C2 data, averaged over the strong IT regions, are also given in Table 4. Mean C2 SLA variance gains are comparable to J2 mission on all IT regions. C2 validation results for M2 IT component show that the ZARON model performs better than other models in most IT regions studied, with a maximum reduction of SSH differences variance of 3.2 cm² on Luzon and 2.2 cm² on Madagascar area. RAY reduces a bit more variance in the Tahiti region; on average over the global ocean, the ZARON and RAY models are the most efficient.


| M2 | | RAY | ZHAO | ZARON | EGBERT | HYCOM | DUSHAW | UBEL |
|---|---|---|---|---|---|---|---|---|
| **Mean variance reduction for J2 database (cm²)** | | | | | | | | |
| SLA | Tahiti | -0.68 | -0.55 | **-0.73** | -0.63 | -0.39 | -0.58 | |
| | Hawaii | -0.65 | -0.58 | **-0.74** | -0.62 | -0.30 | -0.55 | |
| | Madagascar | -0.61 | -0.51 | **-0.68** | -0.66 | -0.10 | -0.41 | |
| | Gulf of Guinea | -0.13 | -0.12 | **-0.14** | -0.10 | -0.02 | -0.05 | |
| | Luzon | -1.37 | -1.22 | **-1.73** | -1.51 | -1.04 | -0.66 | |
| | NATL | -0.15 | -0.13 | -0.18 | -0.16 | -0.08 | -0.09 | **-0.20** |
| | NPAC | -0.29 | -0.28 | -0.35 | -0.30 | -0.13 | -0.25 | **-0.36** |
| | global | -0.23 | -0.20 | **-0.26** | -0.24 | -0.05 | -0.11 | |
| Crossovers | Tahiti | -1.45 | -1.23 | **-1.52** | -1.31 | -0.84 | -1.30 | |
| | Hawaii | -1.93 | -1.92 | **-2.17** | -1.92 | -1.25 | -1.90 | |
| | Madagascar | -0.74 | -0.69 | **-0.79** | -0.81 | +0.50 | -0.45 | |
| | Gulf of Guinea | -0.16 | -0.25 | **-0.26** | -0.12 | -0.05 | -0.24 | |
| | Luzon | -1.83 | -1.75 | **-2.16** | -1.24 | +0.73 | -0.69 | |
| | NATL | -0.11 | -0.11 | -0.09 | -0.09 | +0.25 | +0.09 | **-0.13** |
| | NPAC | -1.05 | -1.01 | **-1.20** | -1.10 | -0.39 | -1.02 | -1.12 |
| | global | **-0.36** | -0.31 | **-0.36** | -0.33 | +0.12 | -0.18 | |
| **Mean variance reduction for C2 database (cm²)** | | | | | | | | |
| SLA | Tahiti | **-0.70** | -0.54 | -0.68 | -0.63 | -0.44 | -0.46 | |
| | Hawaii | -0.56 | -0.47 | **-0.60** | -0.58 | -0.30 | -0.37 | |
| | Madagascar | **-0.55** | -0.45 | **-0.55** | -0.49 | -0.17 | -0.13 | |
| | Gulf of Guinea | -0.09 | -0.07 | **-0.12** | -0.08 | -0.01 | -0.02 | |
| | Luzon | -1.32 | -1.25 | **-1.56** | -1.19 | -1.16 | -0.23 | |
| | NATL | -0.14 | -0.13 | **-0.16** | -0.14 | -0.11 | -0.04 | -0.11 |
| | NPAC | -0.25 | -0.24 | **-0.29** | -0.28 | -0.13 | -0.18 | -0.28 |
| | global | **-0.23** | -0.16 | -0.21 | -0.19 | -0.07 | -0.07 | |
| Crossovers | Tahiti | **-1.78** | -1.27 | -1.68 | -1.42 | -1.28 | -1.17 | |
| | Hawaii | -1.34 | -1.10 | **-1.39** | -1.25 | -0.77 | -0.66 | |
| | Madagascar | -2.08 | -1.55 | **-2.21** | -1.90 | -0.45 | -0.92 | |





| | | | | | | | | |
|---|---|---|---|---|---|---|---|---|
| | Gulf of Guinea | - | - | - | - | - | - | - |
| | Luzon | -3.07 | -2.51 | **-3.22** | -2.39 | -2.61 | -0.80 | |
| | NATL | -0.22 | -0.15 | **-0.24** | -0.20 | -0.14 | +0.02 | -0.11 |
| | NPAC | -0.39 | -0.39 | **-0.47** | -0.42 | -0.12 | -0.29 | -0.45 |
| | global | **-0.60** | -0.45 | **-0.59** | -0.55 | -0.22 | -0.06 | |

**Table 4 : Mean variance reduction for J2 and C2 altimeter databases, within each IT region, when using the different M2 internal tide models and compared to the ZERO correction case; variance reduction of altimeter SLA (white lines) and for altimeters crossovers differences (gray lines) for each mission, in cm². For each IT region, the maximum variance reduction across the different models is in bold.**


**5.2 K1 component**

The maps of K1 SSH variance difference at crossovers using successively the EGBERT, ZARON, and ZHAO IT corrections, are plotted in Figure 11 for the J2 and C2 missions. The K1 IT solutions are compared to a ZERO reference correction. The 3 models have different approaches to take into account the diurnal tides critical latitude and regions where amplitude of K1 IT

is negligible and/or not separable from background ocean variability (cf Sect. 2 and 3.1), which explains the large non defined regions in ZARON and ZHAO maps compared to EGBERT. Results show that the three IT models all reduce the J2 SSH variance significantly in the west Pacific/Luzon and Indonesian regions (more than 2 cm²), while a weak variance reduction is visible in the middle Indian and middle Pacific areas (0.5-1 cm²). The reduction is also important for C2 SSH in the east Pacific/Luzon area and south of Java, and results are noisier in the other oceans where diurnal IT is weak, but C2 data are

likely less efficient for testing K1 tide due its very long alias compared to M2 tide (cf Table 3). The ZARON model reduces slightly more C2 variance in the southern part of the Indian Ocean.

**Figure 11 : Maps of SSH variance differences at crossovers successively using each K1 IT correction and a reference ZERO correction in the SSH calculation for J2 and C2 missions (cm2).**


The maps of SLA variance differences using the EGBERT, ZARON, and ZHAO K1 IT models are plotted in Figure 12 for

the J2 and C2 missions. Spatial SLA patterns are consistent with the SSH maps of Figure 11 and allow a better spatial resolution

compared to SSH maps as also noted for M2 results: using EGBERT model allows a significant reduction of the J2 SLA

variance mostly in the Luzon strait/west Pacific region and the northern Indonesian seas, where the amplitude of the K1 IT is

the most important; a weak variance gain is also visible in the IT regions around Tahiti, Hawaii and north of Madagascar but

also in some large ocean current regions, in the middle Indian ocean and east of Australia. The other maps indicate that ZHAO





is less efficient than the two others in the Luzon region, while ZARON reduces slightly more variance for C2 mission on west Pacific area.

**Figure 12 : Maps of SLA variance differences successively using each K1 IT correction and a reference ZERO correction in the SLA calculation for J2 and C2 missions (cm2).**

The mean statistics of altimeter variance reduction, over the regions defined in Figure 1, are given in Table 5 for the SLA and the SSH differences of J2 and C2 missions and for the different regions studied; notice that we focus on Luzon, Tahiti, Hawaii, Madagascar and global areas because mean K1 statistics are not significant in the other regions of large semidiurnal tides defined in Figure 1. The values in Table 5 indicate a significant variance reduction mainly in the Luzon region as expected




from the analysis of global maps. The ZARON and EGBERT models are the most efficient IT solutions in the Luzon region, with similar variance gains for both models at C2 crossovers. ZARON shows a significant variance gain compared to the ZERO correction for both missions tested, reaching 3 cm² and 2.4 cm² respectively for J2 crossovers and C2 crossovers.


| | K1 | ZHAO | ZARON | EGBERT |
|---|---|---|---|---|
| \multicolumn{2}{c}{**Mean variance reduction for J2 database (cm²)**} | | | |
| SLA | Tahiti | -0.04 | -0.04 | **-0.06** |
| | Hawaii | -0.03 | **-0.05** | **-0.05** |
| | Madagascar | -0.06 | -0.05 | **-0.07** |
| | Luzon | -0.53 | **-1.03** | **-1.09** |
| | global | -0.05 | -0.05 | **-0.06** |
| Crossovers | Tahiti | -0.08 | **-0.14** | -0.10 |
| | Hawaii | -0.05 | **-0.15** | -0.10 |
| | Madagascar | -0.09 | **-0.14** | **-0.14** |
| | Luzon | -1.82 | **-3.01** | -2.85 |
| | global | -0.17 | **-0.21** | -0.12 |
| \multicolumn{2}{c}{**Mean variance reduction for C2 database (cm²)**} | | | |
| SLA | Tahiti | **-0.03** | **-0.03** | **-0.03** |
| | Hawaii | -0.02 | **-0.03** | -0.02 |
| | Madagascar | -0.03 | -0.04 | **-0.05** |
| | Luzon | -0.51 | **-0.86** | -0.80 |
| | global | **-0.04** | **-0.04** | -0.03 |
| Crossovers | Tahiti | -0.02 | -0.04 | **-0.09** |
| | Hawaii | **-0.20** | -0.12 | -0.09 |
| | Madagascar | -0.03 | **-0.07** | -0.04 |
| | Luzon | -1.37 | **-2.41** | **-2.41** |
| | global | **-0.10** | **-0.12** | -0.08 |

**Table 5 : Mean variance reduction for J2 and C2 altimeter databases, within each IT region, when using the different K1 internal tide models and compared to the ZERO correction case; variance reduction of altimeter SLA (white lines) and for altimeters crossovers differences (gray lines) for each mission, in cm². For each IT region, the maximum variance reduction across the different models is highlighted in bold.**





## 6. Wavelength analysis for M2 wave

In order to quantify the impact of each IT model on the altimeter SLA variance reduction as a function of spatial scales, a spectral analysis of J2 along-track SLA is performed. This analysis is not conducted for other missions because the duration of the C2 mission time-series used is too short to allow a proper spectral estimation at the aliasing frequency of M2 (cycle duration is 370 days for C2). Moreover, this diagnostic only focuses on the main M2 IT, because the K1 aliasing frequency by J2 sampling is 173 days (cf. Table 2), which makes it barely separable from the semi-annual ocean signal.

The J2 SLA spectral analysis is performed for each of the IT regions described in Figure 1. For each area, a frequency-wavenumber spectrum is computed for the along-track SLA and for the SLA corrected from each IT solution; the spectral density at 62 days frequency, which is the aliasing frequency band of the M2 tidal component by Jason's orbit, is extracted in both cases and then the normalized difference of the spectral density is computed and plotted as a function of wavelength. This computation gives an estimation of the percentage of energy removed at M2 frequency thanks to each IT model correction, as a function of wavelength and for the different regions studied.

Results for the different regions are gathered in Figure 13 and show that all empirical models generally perform well in removing coherent IT energy for the first mode (wavelengths of about 150 km). Some empirical models also perform well for shorter scales. The DUSHAW model is generally less efficient in the different regions except in the Gulf of Guinea where it is as efficient as others for the first mode. In the Tahiti, Luzon, Gulf of Guinea and NATL regions, ZARON is the most efficient model with a very significant reduction of the energy for the first and the second IT modes: the ZARON model removes 80% of the energy at the M2 frequency for the first internal tide mode and 70% for the second mode in the Tahiti region. With respect to the first mode, the ZARON model removes nearly 80% of the energy in the Gulf of Guinea, 60% in Luzon, Madagascar and NPAC regions, and 50% in the NATL region. We speculate that the regions for which ZARON removes less variance may be regions with stronger IT non-stationarity (Zaron 2017). In the Madagascar region, ZARON, EGBERT, RAY and ZHAO perform similarly for the first mode. Only a few models manage to reduce the IT energy for the second and the third modes: RAY and ZARON reduce more than 60% of the second mode energy around Tahiti and up to 30% in other regions except in NATL where they only reduce about 15 % of the second mode energy. Aside from the fact that models are not perfect, these results corroborate the fact that the non-stationary IT part is even more significant for higher IT modes (Shriver et al. 2014; Rainville and Pinkel 2006). Around Tahiti, the curves indicate that the RAY model also reduces the SLA energy for a third mode of IT (~20%). The ZHAO model also removes some energy at short scales on the Madagascar and Luzon regions.




**Figure 13 : Percentage of the IT signal removed by each IT model as a function of wavelength and for each IT region studied. Blue
line= DUSHAW model, green= EGBERT model, red= HYCOM model, light blue= RAY model, purple=ZARON model, light green=
ZHAO model, black= UBELMANN model.**

The black curves show the performances of the UBELMANN model in the NATL and NPAC regions: it is very efficient in
NPAC with similar energy reduction as ZARON model for the first and second modes and it also removes some signal at
shorter scales. In the NATL area, the UBELMANN model seems to be more efficient than all other models for all wavelengths
and also for large scales, which likely indicates that the model also includes some large scale signals which are not internal
tides but rather some residual barotropic tide signals or even some non-tidal ocean signal aliasing.





The assimilative model, EGBERT, also performs well compared to purely empirical models, but it does not have enough energy for the shorter IT modes except in the Madagascar region where it reduces the SLA energy for scales of 60-70 km.

It is also interesting to point out that the pure hydrodynamic model, HYCOM, removes some energy for the three first IT modes in some of the regions studied: although significantly weaker than for the empirical models, the HYCOM gain reaches 55% for the first mode, 40% for the second mode and 15% for the third mode on Tahiti area. The gain is weak but noticeable in the NATL, NPAC, Luzon and Madagascar regions, but the local rise of energy in some regions also indicates that the hydrodynamic model still has some problems, particularly in the Gulf of Guinea region and for short IT scales in the

Madagascar region.

## 7. Discussion

Seven models of the coherent internal tide surface signature have been extensively compared within the present study: Dushaw 2015; Egbert and Erofeeva 2014; Ray and Zaron 2016; Shriver et al. 2014; Ubelmann, personal communication; Zaron 2019; Zhao et al. 2016. They are of three types: empirical models based upon analysis of existing altimeter missions, an assimilative

model and a three-dimensional hydrodynamic model.

Recently updated Jason-2 and Cryosat-2 altimeter databases have been used to validate these new models of coherent internal tides over the global ocean, focusing on the four main internal tides frequencies, M2, K1, O1, and S2. First, the analysis shows clearly the value of using such a complete altimeter database to validate internal tide models. The great quality of the database allows investigation of small amplitude signals over the entire ocean, and the different sampling characteristics of the various

missions complement each other well. The results point out a significant altimeter variance reduction when using the new internal tide correction models over all ocean regions where internal tides are generating and propagating. Moreover, the spectral approach quantifies the efficiency of the variance reduction potential of each model as a function of horizontal wavelengths—the latter is particularly valuable information for the SWOT mission which will focus as never before on short wavelength phenomena.

All empirical models display generally good performance for M2, K1, O1 and S2, but the DUSHAW solution performs slightly less well. The ZARON and RAY models have similar results for the first three IT modes but the ZARON model removes more variability than all other models over most of the strong IT regions analyzed. It is also noticeable that some models (DUSHAW and ZHAO) still remove some variability in areas of strong currents, likely due to some residual leakage of the mesoscale variability. The UBELMANN solution appears to also remove some large-scale, likely residual barotropic tide

signal, in the north-east of the Azores area.

The assimilative model (EGBERT) performs well relative to the empirical models, but it also removes some variability in regions of strong currents, likely due to some remaining mesoscale variability in the assimilated data.

The hydrodynamic solution, computed from a HYCOM simulation, is also able to reduce some of the internal tide variability in most of the IT regions studied, which is a very encouraging result. However, the analysis indicates that it is not yet mature





enough to be compared to empirical models. The HYCOM solution has significantly stronger amplitudes compared to the other models, which is likely due to the effects of the relatively short HYCOM time series duration (one year) on the IT estimation (see Ansong et al. 2015). Indeed, some tests showed that using a reduction coefficient (Buijsman et al. 2020) that accounts for the short duration of the time series used in the analysis slightly improves the performance of the HYCOM hydrodynamic solution. Ongoing work is testing whether operational HYCOM simulations, which assimilate altimeter

measurements of mesoscale eddies and improve the underlying stratification relative to observations (e.g., Luecke et al. 2017), will yield improvements in the skill of the predicted internal tides in HYCOM.

Following the results presented here, a recommendation has been raised at the last OSTST (Ocean Surface Topography Science Team) meetings of Ponta Delgada Miguel (2018) and Chicago (2019), to use an internal tide model to correct all along track nadir altimeter databases as well as the upcoming high-resolution SWOT swath altimeter missions. Consequently, the Zaron

model is being implemented in the next version of the altimeter GDRs (GDR-F-standard: https://www.aviso.altimetry.fr/data/product-information/updates-and-reprocessing/monomission-data-updates.html), which will be available on AVISO.

In addition, the impact of using the ZARON IT correction has also been estimated for the level-4 (L4) altimeter products, which are global gridded data. A significant improvement was detected in all the regions of interest, and it was demonstrated

that this new correction reduces the remaining IT signal in the L4 AVISO/CMEMS products (Faugère et al., 2019; Zaron and Ray, 2018). Accordingly, this IT correction will be used to compute the SLA for the next DUACS reprocessing product DUACS-2021 which is currently being undertaken. Moreover, the implementation of this new IT correction is planned in the future CMEMS L3 and L4 altimeter product version coming in 2021.

The present study indicates that the use of the altimetry database is a valuable tool to validate models of IT surface signature

on the global ocean and particularly it complements efficiently the in situ validation processes which are generally more localized in space/time due to the availability of in situ datasets (Dushaw et al. 2017, 1995; Zaron and Ray 2017).

Models of the coherent internal tides still need to be improved through inclusion of higher IT modes and more tidal frequencies. In addition, many initiatives are now being conducted to try to better understand and model the non-stationary component of the internal tides. Work is progressing on the modelling of the seasonal and interannual internal tides variability: Zhao (2019),

Zaron (2019), Ray (personal communication), Ubelmann (personal communication). Within the SWOT Science Team and other projects, several teams also work on 3D simulations using different general circulation models such as HYCOM, MITgcm, NEMO (CMEMS-Mercator-Ocean project in progress), or even a specific spectral approach (S. Barbot et al., in preparation).

## 8. Acknowledgements

This work has been performed within the framework of the SWOT-ADT (Algorithm Definition Team) and funded by CNES. We thank R. Baghi for his help on the processings.





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

## 10. Appendices

### Appendix-A- Comparing internal tide models for O1 and S2 waves

Amplitude of O1 and S2 tide components for each IT model are plotted on figures A1 and A2 respectively, on the Luzon area. Concerning O1 tide, Zhao and Zaron show a similar south-west pattern on the west side of the Luzon strait, with an amplitude reaching more than 2 cm for Zaron and only the half for Zhao's solution. On the east side of the strait, the 3 models are quite different: Zhao has the weaker amplitudes, Zaron has strong large scale patterns propagating far eastward (1.5 cm amplitude with 200km-wide features) and decaying to zero above 22°N; and Egbert shows a third very different pattern with zero

amplitude along latitudes 15°N and 21°N, and also east of the Philippines, and amplitudes reaching about 1 cm at 22-23°N. On this region, S2 IT amplitude shows smaller spatial scales than O1, and close to M2 ones as expected. Egbert S2 solution is very different from others and mostly shows a noisy pattern on this Luzon area. Zhao and Zaron show similar features of about 1 cm amplitude and with a clear eastward propagation in the Pacific Ocean and a north-westward direction west of the strait; Zaron has significantly stronger amplitudes.

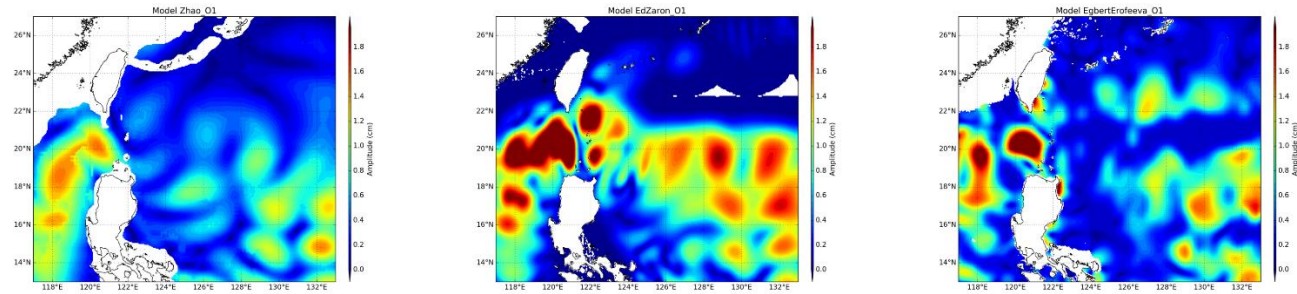

**Figure A1: amplitude of the IT models for O1 tide component on Luzon area**





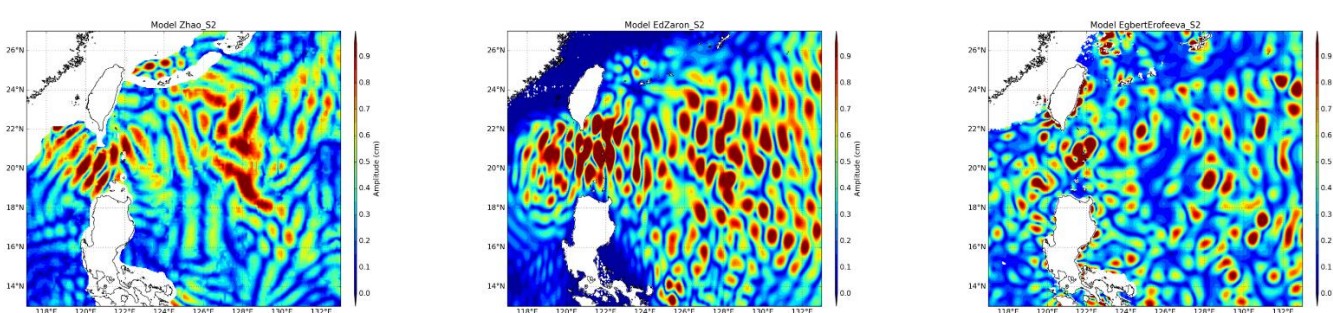

**Figure A2 : amplitude of the IT models for S2 tide component on Luzon area**

Global maps shown on figures A3 and A4 illustrate the mean IT amplitude, and the standard deviation of the IT models for
O1 and S2 tidal component respectively. S2 mean amplitudes show similar patterns as M2 with significantly weaker amplitudes
as expected (below 1 cm); main S2 generation sites are visible around Hawaii in Pacific Ocean, off Amazonia, around
Madagascar, north of Sumatra, south of Lombok, in Banda and Celeb Seas, around Salomon islands, in Luzon area and on
Saipan ridge. O1 IT has similar patterns as K1 with significantly weaker mean amplitudes.

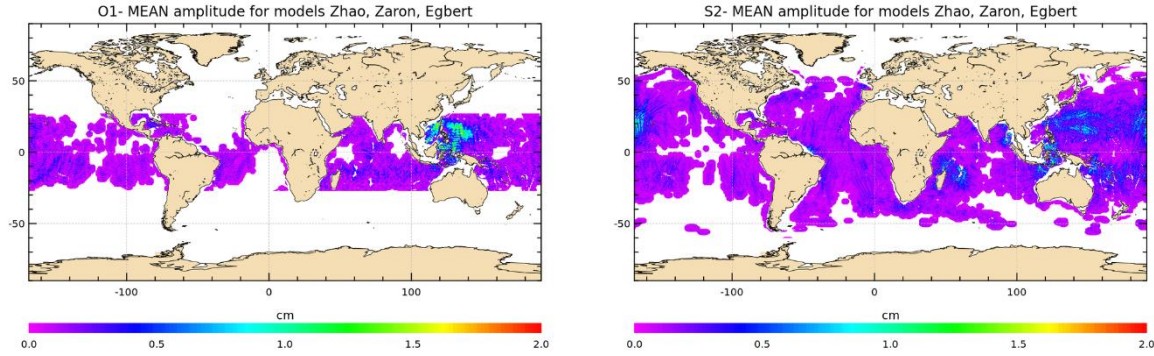


**Figure A3 : global maps of mean amplitude of the O1 and S2 IT models (cm)**

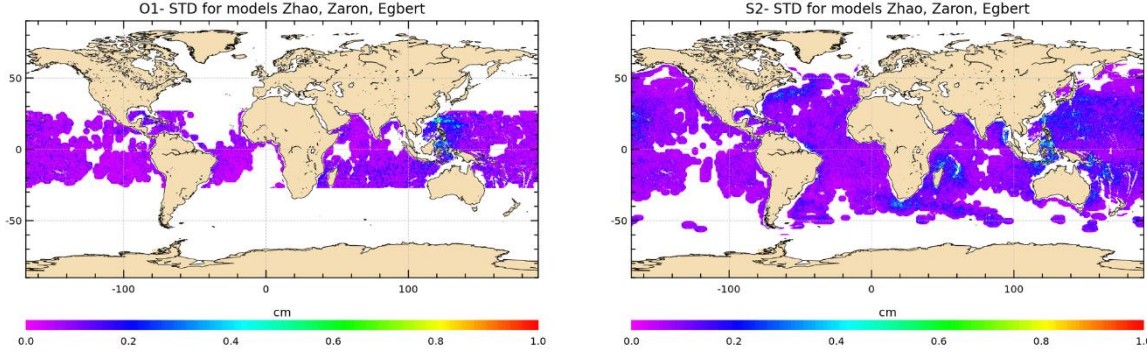

**Figure A4 : global maps of standard deviation of the O1 and S2 IT models (cm)**





For O1, the Luzon strait region mainly stands out with stronger standard deviation values on Luzon strait and eastern in the Philippines sea (values around 0.4 cm). S2 standard deviation reaches 0.1-0.5 cm in the Hawaii, Madagascar and Luzon regions where the amplitude of S2 IT signal is the more important. The mean standard deviation is computed on the different regions studied, using only data located in deep ocean and values are gathered in Table A5. S2 mean standard deviation is at least 3 times smaller compared to M2, which is coherent with the fact that S2 IT has significantly smaller amplitude; stronger values

occur on Luzon and Madagascar regions where mean S2 IT amplitude is maximum. O1 IT has the strongest standard deviation (0.18 cm) in the same Luzon area as K1, where diurnal internal tides have the most significant amplitudes in the ocean, which indicates that O1 IT models have some uncertainties in this region.

| STD for deep ocean (cm) | S2 | O1 |
|---|---|---|
| Tahiti | 0.08 | 0.06 |
| Hawaii | 0.11 | 0.06 |
| Madagascar | 0.15 | 0.08 |
| Gulf of Guinea | 0.08 | 0.06 |
| Luzon | 0.16 | 0.18 |

**Table A5: mean standard deviation of models for S2 and O1 tide components for each studied region (in cm)**


**Appendix-B- Validation results for O1 internal tide models**

The maps of SLA and crossovers variances differences using each of the three different O1 IT models are plotted on figures B1 and B2 resp., for both J2 and C2 missions; the O1 IT solutions are compared to a ZERO reference correction. First, it is noticeable that as for K1, the ZARON O1 solution is not defined on large ocean regions mostly taking into account the diurnal

critical latitude and regions where O1 IT amplitude is negligible and/or not separable from background ocean variability. The ZHAO O1 solution is not defined beyond the diurnal tide critical latitude, while EGBERT solution is defined on a wider range of latitudes.

The three models remove a significant amount of J2 SLA variance mostly in the Luzon strait/west Pacific region where the amplitude of the O1 IT is the most important in the ocean, the variance reduction reaches 1-2 cm² in this area. EGBERT model

removes some C2 variability (0.5 cm² on C2 SLA) in the middle of Indian ocean around latitude 20°S, but maps are noisier for the 2 other models in this region; some C2 SLA variance reduction occurs west of Luzon strait and north of Indonesians seas, but in the Philippines sea the three models both reduce and raise the C2 SLA variance on the 10°-25°N latitude band with a zonal band pattern; the variance raise is minimum with EGBERT model. This zonal effect only visible on C2 SLA data might be explained by some residual TP-Jason errors or even oceanic variability in the O1 IT models in this area. The maps of the




variance differences at crossovers are consistent with SLA results for both missions, and they indicate a significant J2 variance reduction in the Indonesian and Philippines areas; the C2 crossovers maps indicate a weaker and noisier impact compared to J2 data.

**Figure B1 : Maps of SLA variance differences successively using each O1 IT correction and a reference ZERO correction in the**
**SLA calculation for J2 and C2 missions (cm2).**





**Figure B2 : Maps of SSH variance differences at crossovers successively using each O1 IT correction and a reference ZERO correction in the SSH calculation for J2 and C2 missions (cm2).**

The mean statistics of altimeter variance reduction for O1 IT are given on table B3 for the SLA and the SSH crossovers differences of J2 and C2 missions; notice that only Luzon region is presented because O1 amplitude is not significant elsewhere. The figures show that ZARON O1 model reduces more J2 variance than other models (1.5 cm² of SSH crossover variance on the area), but EGBERT and ZHAO solutions are a bit more efficient when considering mean C2 SLA values. Mean C2 crossovers variance differences are very weak, reflecting the noisy corresponding variance maps in the region as

seen on figure B1 and figure B2. These weaker/noisier results noted with C2 crossovers for O1 frequencies can likely be



explained by the fact that the C2 temporal series are shorter than J2 ones which make the analysis noisier particularly for such small amplitude signal, in addition to the fact that crossovers statistics are smoothed on larger boxes compared to SLAs.

| Mean variance reduction on Luzon region | ZHAO | ZARON | EGBERT |
|---|---|---|---|
| **Mean variance reduction for J2 database (cm²)** | | | |
| SLA | -0.30 | **-0.41** | **-0.41** |
| crossovers | -1.15 | **-1.53** | -1.14 |
| **Mean variance reduction for C2 database (cm²)** | | | |
| SLA | -0.35 | -0.13 | **-0.46** |
| crossovers | **-0.18** | -0.11 | -0.12 |

**Table B3: Mean variance reduction for J2 and C2 altimeter databases, in the Luzon region, when using the different O1 internal tide models and compared to the ZERO correction case; variance reduction of altimeter SLA (white lines) and for altimeters crossovers differences (gray lines) for each mission, in cm². The maximum variance reduction across the different models is highlighted in bold**

**Appendix-C- Validation results for S2 internal tide models**

The maps of SLA and crossovers variance differences using each of the three different S2 IT models are plotted on figure C1 and C2 resp., for both J2 and C2 missions; the S2 IT solutions are compared to a ZERO reference correction. The ZARON S2 solution is not defined on large deep ocean regions (white areas on the maps) where S2 IT amplitude is negligible and/or not separable from background ocean variability.

Using the three models for S2 IT correction allows a small but well-detected reduction of the J2 and C2 SLA variances in the
same regions of the ocean as for the main semi-diurnal IT (cf Figures 7-10 for M2 IT): variance reduction is maximum (about 0.5-1cm²) west of Hawaii region, north of Madagascar, in the Luzon strait/west Pacific region, and also in the Indonesian islands, north of Sumatra and between Salomon islands and New Zealand. The C2 maps show similar reduction patterns but the variance gain is weaker than for J2. Both EGBERT and ZHAO models remove some variance south of Africa in the Agulhas currents while ZARON does not; ZHAO model clearly impacts the altimeter variance in  most of the great ocean
currents areas, which likely indicates that the model might contain some residual oceanic signal and/or some J2 error and not only IT. The patterns of the crossover variance differences are consistent for J2 missions but with weaker values than for the SLA; for C2 mission, the crossovers maps indicate a weaker and noisier impact than for SLA as already noted for O1 frequency.





**Figure C1 : Maps of SLA variance differences successively using each S2 IT correction and a reference ZERO correction in the SLA**
**calculation for J2 and C2 missions (cm2).**







**Figure C2: Maps of SSH variance differences at crossovers successively using each S2 IT correction and a reference ZERO correction in the SSH calculation for J2 and C2 missions (cm2).**

The mean statistics of the altimeter variance reduction are gathered in table C3 for the SLA and the SSH differences of the J2 and C2 missions and for the different regions studied; the analysis focuses on Tahiti, Hawaii, NPAC, Madagascar, and Luzon areas because mean S2 statistics are not significant elsewhere. The figures show weak SLA variance reductions with stronger values on the Luzon, Madagascar and the Hawaii regions where amplitude of the S2 IT is the most important: if looking at J2 SLA, the 3 models are equivalent on Hawaii and Madagascar, but EGBERT and ZARON are more efficient to reduce variance

on Luzon area (0.28 cm² and 0.25 cm² resp.). Looking at C2 SLA, the 3 models give similar results on Madagascar, EGBERT




and ZHAO reduce more variance on Luzon region and EGBERT is more efficient on Hawaii region. Unlike the results obtained for the M2 and K1 waves and described in previous sections, the variance reduction for crossovers differences is weaker than for the SLA for S2 wave and mean values are hardly useful; this is likely explained by the weak S2 IT signal in the ocean in addition to the fact that crossover statistics are performed on large boxes which tends to smooth it even more. This analysis
results suggest that EGBERT and ZARON S2 IT solutions are the most efficient on the different regions of interest.

| | Mean variance reduction (cm²) | ZHAO | ZARON | EGBERT |
|---|---|---|---|---|
| | **Mean variance reduction for J2 database (cm²)** | | | |
| SLA | Tahiti | -0.02 | -0.02 | **-0.05** |
| | Hawaii | -0.14 | -0.15 | **-0.17** |
| | Madagascar | -0.14 | -0.14 | **-0.15** |
| | Luzon | -0.15 | -0.25 | **-0.28** |
| | NPAC | -0.05 | -0.06 | **-0.07** |
| | global | -0.04 | -0.04 | **-0.06** |
| Crossovers | Tahiti | -0.04 | -0.04 | **-0.05** |
| | Hawaii | **-0.12** | -0.11 | -0.11 |
| | Madagascar | -0.02 | -0.02 | **-0.03** |
| | Luzon | 0 | -0.01 | **-0.05** |
| | NPAC | **-0.07** | -0.06 | **-0.07** |
| | global | 0 | **-0.02** | **-0.02** |
| | **Mean variance reduction for C2 database** | | | |
| SLA | Tahiti | -0.01 | -0.01 | -0.02 |
| | Hawaii | -0.09 | -0.08 | **-0.15** |
| | Madagascar | -0.08 | -0.06 | **-0.09** |
| | Luzon | **-0.16** | -0.09 | **-0.16** |
| | NPAC | -0.04 | -0.04 | **-0.07** |
| | global | -0.01 | **-0.02** | **-0.02** |
| | | | | |
| Crossovers | Tahiti | 0 | 0 | +0.01 |
| | Hawaii | **-0.01** | **-0.01** | 0 |
| | Madagascar | **-0.02** | +0.02 | 0 |
| | Luzon | +0.03 | +0.02 | +0.02 |
| | NPAC | 0 | 0 | 0 |
| | global | 0 | 0 | 0 |





**Table C3 : Mean variance reduction for J2 and C2 altimeter databases, within each IT region, when using the different S2 internal tide models and compared to the ZERO correction case; variance reduction of altimeter SLA (white lines) and for altimeters crossovers differences (gray lines) for each mission, in cm² (0 is for |value| < 0.005 cm²). For each IT region, the maximum variance reduction across the different models is highlighted in bold**
