# Peer review of "Accuracy assessment of global internal tide models using satellite altimetry"

_Ocean Science, 2020_

## Referee Comment (RC1) · C. K. Shum (Referee) · 26 Jul 2020

Reviewers: Jason J. Otero-Torres (lead); C.K. Shum

General Comments:

This was a very detailed assessment and good comparison of available IT models supported by qualitative, quantitative and spectral analyses. The manuscript is in overall well structured, clearly written and organized. Starting from the abstract, all sections in the manuscript were captured concisely. Next, the introduction paragraph provided a summarized background on satellite altimetry and stated its limitations for tidal analyses, supported by relevant references. Also, it explains the need for a validation process for available IT models and delineates what to expect in subsequent sections

but omits some information on the resolution of altimetry data used. Next, in the presentation of the participating IT models' section, all the participating models were well described in terms of their methodology, except for Ray model. Next, the qualitative and quantitative comparison sections are well presented, requiring minor grammar corrections. All the plots are accurately described, but some plots were difficult to read due to their font size. Also, mathematical support or presentation of equations used (as cited in the literature), are rather necessary in the quantitative comparison section. Next, the presentation of the altimeter database section is well written and clearly indicate the reasons for selecting the J2 and C2 missions for the comparison. Next, the method of comparison section was well explained and follows a logical order but needs for a clearer or possibly an enumerated sequence. In the next section, variance reduction analysis using satellite altimeter data, SSH and SLA variances plots were produced for both the M2 and K1 constituents. The results were properly described, and variances of the corrections were stated for each participating IT model. Some plots were difficult to read due to their font size. Next, the wavelength analysis of M2 section corroborates previous results by estimating the amount of energy removed at from participating IT models. Results for this section are well described; only need to further elaborate on EGBERT's model performance in the Gulf of Guinea. Next, the discussion session clearly summarizes previous sections and offers conclusive explanations on the performance of each model based on final results. Finally, the appendices section supplemented the manuscript, with results of remaining constituents. This section was clearly described and presented similarly to the variance reduction analysis of M2 and K1 section. As previous sections, the font size on some plots were not legible.

Specific Comments:

1. Line 83: Remove "Altika". The mission is not used for the quantitative comparison or anywhere in the manuscript.

2. Line 84: The section omits the resolution of the altimetry databases used in the validation process e.g. HRM/LRM

3. Lines 135-144: RAY model needs to be better described in the presentation of participating internal tide models. The model methodology should be included as for other participating models or more information is rather necessary.

4. Lines 240-243: Consider mentioning why from all the seven regions of interest, NPAC and Luzon regions were selected for the comparison e.g. more energetic regions of all seven.

5. Line 294: Include Stammer, 2014 equations for the calculation of STD of IT models for the reader convenience.

6. Line 325: Here the resolution of 1-hz is indicated (LRM). Consider adding it to line 84.

7. Line 351: For future research: consider adding seasonal barotropic tide correction in best performing models.

8. Lines 371-408: This section explains well the methodology for the analysis. But would be convenient to enumerate each step to follow a sequential order.

9. Lines 416-425: Reiterate or remind that the quantification and regional impact of the M2 IT correction were performed using all participating IT models but not the same case for K1.

10. Line 450: Independent results from C2 shows similar patterns as J2 mission, albeit J2 bias in empirical IT models. From all IT models listed in table 1, only ZARON incorporates Altika mission. Consider adding Altika to support C2 independent results and further corroborate J2 bias towards empirical model.

11. Line 419-460: Speculate or provide a possible explanation of why Dushaw and HY-COM models rise SSH or SLA variances in some locations as supported by conclusive evidence in the variance computation e.g. areas of strong currents, others?

12. Line 559: Further elaborate on EGBERT model's performance in the Gulf of

Guinea. EGBERT model appears to reduce energy in shorter modes for this region, compared to other models.

Technical Corrections:

1. Line 32: add colon: "tidal constituents: M2…"

2. Line 33-34: …Cryosphere Satellite-2 data. (Remove) "taking advantage of the long-term altimeter databases available"

3. Line 84: …or C2 hereafter). (Remove) "taking advantage of the long-term altimeter databases available"

4. Figure 2: Capitalize first letter "amplitude"

5. Figure 3: Capitalize first letter "amplitude"

6. Figure 4: Capitalize first letter "amplitude", replace "on" with "in"

7. Line 32: include colon after constituents":"

8. Line 84: state the resolution mode of the altimeter databases selected: LRM/ HRM

9. Line 299: re-write sentence: …Figure 5 and 6 (remove) "respectively" the mean (add) "amplitudes" and the standard deviation of the M2 and K1 models (add) "respectively".

10. Line 304: Capitalize first letter "oceans"

11. Line 305: abbreviate standard deviation (STD)

12. Line 306: abbreviate standard deviation (STD)

13. Line 308: Tahiti and Hawaii, and (remove) "also" in the Madagascar region.

14. Line 351: lower cap "Tide", lower cap "Other"

15. Figure 5: Capitalize first letter "global"

16. Figure 6: Capitalize first letter "global"

17. Table 2: re-write headers: Region, STD M2 (cm), STD K1(cm). Abbreviate standard deviation (STD)

18. Line 343: word of choice: "most difficult" instead of "trickiest". Abbreviate Jason (J2)

19. Line 371: (Remove) "The first step consists in generating" (add) First we generate the corresponding. . .

20. Line 375: Second, the altimeter. . .

21. Line 379: Third, the altimeter. . .

22. Line 383-388: re-write entire paragraph to connect with previous

23. Line 386: (remove) "As" (add) Since coherent IT have. . .

24. Line 390: Fourth, the maps. . .

25. Line 397: For J2, all latitudes are covered bu the number of points is are limited. . .

26. Line 398: equator and increase towards. . . .

27. Line 399: Fifth, the mean. . .

28. Figure 7: Use larger font for bottom left plot. In the figure text, use exponent "(cm2)"

29. Figure 8: Correct plot top left plot "RRAY" to RAY. Use larger font for bottom left plot. In the figure text, use exponent "(cm2)"

30. Figure 9: Correct plot top left plot "RRAY" to RAY. Use larger font for bottom left plot. In the figure text, use exponent "(cm2)"

31. Figure 10: Use larger font for bottom left plot. In the figure text, use exponent "(cm2)"

32. Figure 11: In the figure text, use exponent "(cm2)"

33. Figure 12: In the figure text, use exponent "(cm2)"

34. Line 540: add comma (Zaron, 2017)

35. Line 567: Abbreviate internal tide (IT)

36. Line 572: Abbreviate internal tide (IT)

37. Line 576: Abbreviate internal tide (IT)

38. Line 581: add coma: IT modes, but. . .

39. Figure A1: Capitalize first letter "amplitude"

40. Figure A2: Capitalize first letter "amplitude"

41. Figure A3: Capitalize first letter "global"

42. Figure A4: Capitalize first letter "global"

43. Table A5: Capitalize first letter "mean"

44. Line 786: Abbreviate internal tide (IT)

45. Line 788: B1 and B2 respectively

46. Figure B1: In the figure text, use exponent "(cm2)"

47. Figure B2: In the figure text, use exponent "(cm2)"

48. Line 824: Abbreviate internal tide (IT)

49. Figure C1: In the figure text, use exponent "(cm2)"

50. Figure C2: In the figure text, use exponent "(cm2)"

51. Figure C3: add period "."

Please also note the supplement to this comment:
https://os.copernicus.org/preprints/os-2020-57/os-2020-57-RC1-supplement.zip

---

## Referee Comment (RC2) · Anonymous Referee #2 · 30 Jul 2020

In this work the authors assess the accuracy of 7 internal tide models and their ability to for correcting satellite altimetry data sets. Three different types of models are included: empirical, assimilative and hydrodynamic. The models are compared amongst each other in a qualitative and a quantitive approach; then their ability to reduce variance of SSH and SLA from satellite altimetry is assessed.

General comments:

The manuscript is well-written, laid out clearly, and is easily readable. It could benefit from some improvements in the grammar used.

The figure labels are hard to read as they are very small and some information which is repeated on every figure panel could be omitted and/or moved to e.g. the caption.

[Figure]

No method is numerically explained using an equation/equations – the authors should consider adding these for clarity and for ease of anyone else wanting to repeat or carry out a similar analysis.

Detailed comments:

L136: How is this map constructed?

L240: Why do you specifically select these two regions for the comparison?

L294: Does a simple standard deviation give you the most robust measure of variation between the models? Would you not expect larger variations in areas with larger amplitudes? What about including a measure of e.g. STD normalised by the mean amplitude?

L296: Why does the DUSHAW increase the STD so much?

Lines 333-357: What time periods do the two datasets span?

L390: What resolution do the JS and CS tracks have?

Table 4: check that the highlighted values really correspond to the best reduction. E.g. for J2, crossover, Madagascar EGBERT gives the best reduction, not ZARON.

Figure 13: In this figure the caption (percentage of IT signal removed) does not correspond to the y-axis label (ratio of power spectral density (cmˆ2.km)

Figures general:

- You tend to use the same color bars for all subplots in your images. You could plot one large colorbar at the bottom with labels that have a bigger font size. The resulting white space could be used to make the plot titles larger (see next comment).

- Your subplot titles include information that is repeated multiple times – e.g. in Fig. 7 all subplots have 'Mission j2, cycles. . .' – could this go in the caption? Make the plot headers larger as they are not legible at 100% size.

[Figure]

Technical comments:

L49: at -> et

L57: coming -> upcoming

L88: proposed -> presented

L108: fit -> fitted

L109-110: grammar

Table 2: use consistent notation (comma or dot)

L296: notice -> note

L375: The altimeter SSH using successively each of the IT corrections tested → The altimeter SSH using IT corrections from each model, resepectively, . . . (successively is used in a confusing way more than once in the document – check the other occurrences)

L432: ZHAO model -> the ZHAO model

L433: four models, RAY -> four models RAY,. . .

L450: notice -> note

Figure 9a: RRAY -> RAY

---

## Short Comment (SC1) · 11 Aug 2020

Review of: "Accuracy assessment of global internal tide models using satellite altimetry" By: Loren Carrere, et al Reviewed by: Chris Unsworth, Jenny Jardine, Marta Payo Payo & anonymous Institution: National Oceanography Centre, Liverpool, UK

Description: Carrere et al., present a timely and desired analysis of a range of methods to correct satellite sea level altimetry data for the effects of internal tides. The effects of internal tides on satellite altimetry data are well known to be aliased due to satellite return periods being greater than the dominant periods of internal tides. Carrere et al., select 7 regions of the Earth to do their analysis – notable for their lack of seasonal stratification and the presence of internal tides. Comparisons are made from the per-

spective of variance reduction - as no pure "truth" measurement is possible - the only way to know if better results are made is by comparison. Comparisons are made for "free" models, and models with data assimilation. However, it should be noted that some of the data used in the assimilation is also the data used for the comparison – so it is perhaps unsurprising that these show the least variance.

Main Comments: Two other reviewers have picked up typos and inconsistencies, but both say the manuscript is clear. The majority of our reviewers found the manuscript very confusing. Our comments are summarised below.

1. There is a considerable amount of assumed knowledge in the manuscript. It took us a long time to work out that you were trying to use the models to correct the altimetry data. The abstract is especially hard to understand – this needs to be as clear as possible! Try getting a colleague that doesn't know anything about the subject to read it and see if they can work out the purpose, key results and implications of the paper.

2. Although the tables and figures (once font sizes have been made large enough to read) communicate the results well – the method of getting the data in the tables is highly unclear. No equations are presented and no description of how comparisons are made across models is given – as the different models may have different resolutions, time stepping and may have been run for different lengths of time it is hard to know if you have made "like for like" comparisons. A separate table, or additional columns in table 1, would be very useful to understand exactly what you have done.

3. There is a considerable amount of vague and non-scientific terminology used when describing the results, in the discussion and in the conclusions. Phrases and words such as "performs well", "some", "weaker", "some problems" are very unscientific ways of describe your results – especially considering the manuscript focuses on validation and numerical comparisons of models. If you have the numbers to quantify these statements you use them at all times. The use of the word "significant" is very misleading. No statistical test is described to give that word a scientific meaning. If you have performed a significance tests on your data you need to detail that because it would help you communicate your results more clearly and forcefully, if not – you need a different word, or a quantification (e.g. a fraction or percentage).

4. Your objectives seem to be incorrect or are misleading. Objective one is stated as: "The objective of this paper is to present a detailed comparison and a validation assessment of these internal tide models using satellite altimetry". As far as we can tell, this is untrue. No validation of the models themselves is provided. You are comparing the internal tide models, but you cannot validate the models per se- as no "truth" measurement is possible. We found this objective confusing – as we tried to find where you had validated your models. Objective 2: "The analysis focuses on the correction of the satellites' measurements from the coherent internal tide signal for the main tidal constituents, M2, S2, K1 and O1". This objective seems more correct based on the results presented.

We agree with reviewer #2 that the font in the figures needs to be larger, at least 2x as large. We also suggest using a different colour scheme - one that colour-blind people can use. Colourmaps that do this are common on the internet, some examples are given below:

https://colorbrewer2.org/#type=sequential&scheme=BuGn&n=3

https://www.scientificamerican.com/article/end-of-the-rainbow-new-map-scale-is-more-readable-by-people-who-are-color-blind/

The manuscript does present highly relevant work that the community would like to see, however its communication is poor. As it does not appear that re-working of the models or analysis is needed, we think the study needs minor revisions; but the authors need to take care how they communicate their work.

Minor Comments: Abstract: 1) One sentence does not make a paragraph (this error is present throughout the manuscript)

2) "In order to access the targeted ocean signal..." what is the targeted ocean signal? This phrase is not used anywhere else in the manuscript and it is not explained. Start with satellite altimetry- and why it has a source of error due to internal tides, and why correcting it is important (in ~2-3 sentences).

3) "several geophysical parameters" – the list of geophysical parameters that exist on Earth is enough to fill a book – what do you mean exactly?

4) I would also include that this work led to the Zaron model being implemented in the GDR standard (if this is true)

Methods You need to describe your models more consistently. At present the level of detail given to these models is sporadic. What years are they all run for? At what resolutions? What time-stepping? This kind of information is important as it is relevant for how you compare the results. Description how you compare the data is also needed, equations may well help you communicate this clearly. Are any statistical tests used? How many degrees of freedom are allowed when you calculate the variance?

Discussion The paragraph startling line 597: "Following the results presented here, a recommendation.." This is very interesting, but it unclearly worded. You say the results presented here (as in now) led to the Zaron model being implemented in the GDR standard, which was decided a couple of years ago. This is very significant and important. You could reframe the manuscript as the scientific justification for that decision (which has obviously been presented at conferences prior to submission to the journal – as is usually done). I would include this decision, and the implications of the increase in capability, in the introduction as well

Break these long sentences up. E.g. "The present study indicates that the use of the altimetry database is a valuable tool to validate models of IT surface signature on the global ocean and particularly it complements efficiently the in situ validation processes which are generally more localized in space/time due to the availability of in situ datasets (Dushaw et al. 2017, 1995; Zaron and Ray 2017)."

Turns into: "The present study indicates that the use of the altimetry database is a valuable tool to validate models of IT surface signature on the global ocean. It particularly complements the in-situ validation processes which are generally more localized in space/time due to the availability of in situ datasets (Dushaw et al. 2017, 1995; Zaron and Ray 2017)."

A general rule to help is: once sentence has one point, or message. If you have two messages, you need two sentences. This shorter and more direct structure especially helps for non-native-English readers.

"In addition, many initiatives are now being conducted to try to better understand and model the non-stationary component of the internal tides. Work is progressing on the modelling of the seasonal and interannual internal tides variability: Zhao (2019), Zaron (2019), Ray (personal communication), Ubelmann (personal communication). Within the SWOT Science Team and other projects, several teams also work on 3D simulations using different general circulation models such as HYCOM, MITgcm, NEMO (CMEMS-Mercator-Ocean project in progress), or even a specific spectral approach (S. Barbot et al., in preparation)."

We can understand why you want to say this, but none of this is produced in the manuscript you present, so it is not a conclusion you are able to make. You could discuss these ongoing efforts in the light of your new findings, and the implications of the present work has on these efforts. But it is not a conclusion.

---

## Editor Comment (EC1) · Mattias Green (Editor) · 21 Aug 2020

The authors compare and assess the performance of a suite of existing models of the global internal tides to each other. The methods are well established and this is a paper that I feel should be published, especially in light of recent review papers about tides. There are comments from two reviewers and one short note that will all require replies and amendments to the manuscript. I agree with these, and I have a few other comments on my own (mainly on the presentation of the figures) below.

Minor comments "Significantly" implies statistics has been done to show this – please reword if you mean large or major. Colour figures: The colourmaps used are not 100% friendly for those with impaired colour vision. I suggest they are remade using a

[Figure]

suitable cmocean map instead. At the same time, make sure the text in the figures is readable. Table 2: M2 should have a dot as decimal placeholder, not comma. Figure 7: please use a diverging colour scheme for difference plots, e.g., blue-white-red. Please mark the regional maps on the large maps and tell us in the caption where they are. Sometimes you write Jason-2, sometimes J2. Please stick to one; I prefer the full name spelled our for all the satellites myself since there is a J2 tidal constituent.

---

## Author Comment (AC1) · 17 Sep 2020

**Response to Referee #1 :**

Dear C. K. Shum,

Thank you for your interest in this manuscript and for the comments and suggestions you make.

I reply to all your comments, corrections and suggestions to change hereafter in blue.

Best regards,

Loren Carrere

**General comments:**

This was a very detailed assessment and good comparison of available IT models supported by qualitative, quantitative and spectral analyses. The manuscript is in overall well structured, clearly written and organized. Starting from the abstract, all sections in the manuscript were captured concisely. Next, the introduction paragraph provided a summarized background on satellite altimetry and stated its limitations for tidal analyses, supported by relevant references. Also, it explains the need for a validation process for available IT models and delineates what to expect in subsequent sections but omits some information on the resolution of altimetry data used. Next, in the presentation of the participating IT models' section, all the participating models were well described in terms of their methodology, except for Ray model. Next, the qualitative and quantitative comparison sections are well presented, requiring minor grammar corrections. All the plots are accurately described, but some plots were difficult to read due to their font size. Also, mathematical support or presentation of equations used (as cited in the literature), are rather necessary in the quantitative comparison section. Next, the presentation of the altimeter database section is well written and clearly indicate the reasons for selecting the J2 and C2 missions for the comparison. Next, the method of comparison section was well explained and follows a logical order but needs for a clearer or possibly an enumerated sequence. In the next section, variance reduction analysis using satellite altimeter data, SSH and SLA variances plots were produced for both the M2 and K1 constituents. The results were properly described, and variances of the corrections were stated for each participating IT model. Some plots were difficult to read due to their font size. Next, the wavelength analysis of M2 section corroborates previous results by estimating the amount of energy removed at from participating IT models. Results for this section are well described; only need to further elaborate on EGBERT's model performance in the Gulf of Guinea. Next, the discussion session clearly summarizes previous sections and offers conclusive explanations on the performance of each model based on final results. Finally, the appendices section supplemented the manuscript, with results of remaining constituents. This section was clearly described and presented similarly to the variance reduction analysis of M2 and K1 section. As previous sections, the font size on some plots were not legible

**Specific comments:**

1. Line 83: Remove "Altika". The mission is not used for the quantitative comparison or anywhere in the manuscript. LC: OK removed.

2. Line 84: The section omits the resolution of the altimetry databases used in the validation process e.g. HRM/LRM - LC : LRM added

3. Lines 135-144: RAY model needs to be better described in the presentation of participating internal tide models. The model methodology should be included as for other participating models or more information is rather necessary. LC: added reference to paper Ray and Zaron 2016 section3 + a few sentences on methodology

4. Lines 240-243: Consider mentioning why from all the seven regions of interest, NPAC and Luzon regions were selected for the comparison e.g. more energetic regions of all seven. LC : they are more energetic regions + all tested models are available on NPAC region and Luzon area is characterized by strong semi-diurnal and diurnal baroclinic tides. Information added in the text.

5. Line 294: Include Stammer, 2014 equations for the calculation of STD of IT models for the reader convenience. LC: done

6. Line 325: Here the resolution of 1-hz is indicated (LRM). Consider adding it to line 84. : LC: OK

7. Line 351: For future research: consider adding seasonal barotropic tide correction in best performing models. LC: I agree that this point could be an interesting point to notice, but as this correction is not yet available and used in the present dataset, I think that mentioning it might make the definition of the SSH a bit confusing.

8. Lines 371-408: This section explains well the methodology for the analysis. But would be convenient to enumerate each step to follow a sequential order. LC: OK modified in the text.

9. Lines 416-425: Reiterate or remind that the quantification and regional impact of the M2 IT correction were performed using all participating IT models but not the same case for K1. LC: OK added in the text in sections M2 and K1.

10. Line 450: Independent results from C2 shows similar patterns as J2 mission, albeit J2 bias in empirical IT models. From all IT models listed in table 1, only ZARON incorporates Altika mission. Consider adding Altika to support C2 independent results and further corroborate J2 bias towards empirical model. LC: Indeed the tests with Altika mission have also been performed and presented at some conferences (OSTST), but for the clarity of the paper we prefered not to include it. Moreover analysis with Altika mission gives close results to C2 and J2 tests.

11. Line 419-460: Speculate or provide a possible explanation of why Dushaw and HYCOM models rise SSH or SLA variances in some locations as supported by conclusive evidence in the variance computation e.g. areas of strong currents, others? LC: some comments have been added in the text:

The DUSHAW model raises SLA variance in several mesoscale regions (Gulf Stream, Agulhas current, Malvinas region and Kuroshio currents), indicating that the model does not properly separate IT and other ocean signals in these strong current areas.

HYCOM raises the variance over wider regions in the three oceans than the empirical and assimilative models do, likely due to its intrinsic characteristic of free hydrodynamic model which may induce more phase errors compared to constrained/empirical models + due to the short HYCOM time series duration used to extract the IT atlas and that induces stronger IT amplitudes (see Ansong et al. 2015 and Buijsman 2020)

12. Line 559: Further elaborate on EGBERT model's performance in the Gulf of Guinea. EGBERT model appears to reduce energy in shorter modes for this region, compared to other models.
LC: comment added

**Technical corrections:**

LC : All technical corrections proposed have been taken into account and the font size has been enlarged on the plots to be more legible.

---

## Author Comment (AC2) · 17 Sep 2020

**Response to Anonymous Referee #2 :**

Dear Referee,

Thank you for your interest in this manuscript and for the comments and suggestions you make.

I reply to all your comments and suggestions to change hereafter in blue.

Best regards,

Loren Carrere

**General comments:**

No method is numerically explained using an equation/equations – the authors should consider adding these for clarity and for ease of anyone else wanting to repeat or carry out a similar analysis.

LC: the equations used has been added in section 3.2 and also in section 4.2.

**Detailed comments:**

L136: How is this map constructed?

LC: the complete description of the method used is available in the paper Ray and Zaron 2016 (section 3 for the construction of the empirical maps): this reference and a few more sentences have been added in the text.

L240: Why do you specifically select these two regions for the comparison?

LC: These 2 regions are the more energetic regions of all the seven considered. Moreover, all 7 models are available on NPAC region, and the amplitude of baroclinic tides is important both for M2 and diurnal tides on LUZON. Info added in the text.

L294: Does a simple standard deviation give you the most robust measure of variation between the models? Would you not expect larger variations in areas with larger amplitudes? What about including a measure of e.g. STD normalised by the mean amplitude?

LC: I have also computed the STD normalised by the mean amplitude (cf figure below), but values become very big in large regions due to the fact that amplitude of IT is very weak in many places … generally, the value of this ratio is about 0.2-0.3 around IT generation regions and some clear beams patterns where models agree with each other are detected. I've added a comment on the value of this ratio in the text: line 329-330.

[Figure]

[Figure]

L296: Why does the DUSHAW increase the STD so much?

LC: because DUSHAW's maps are noisier on wider regions and likely include some more different patterns than other models and also locally greater phase differences. DUSHAW model also includes some discontinuities between areas used to compute the global solution. Some comments have been added in the text.

Lines 333-357: What time periods do the two datasets span?

LC: information added in table 3

L390: What resolution do the JS and CS tracks have?

LC: both are 1-Hz along-track measurements = LRM. The information is added in introduction + section 4.1 which describes the data.

Table 4: check that the highlighted values really correspond to the best reduction. E.g. for J2, crossover, Madagascar EGBERT gives the best reduction, not ZARON.

LC: corrected

Figure 13: In this figure the caption (percentage of IT signal removed) does not correspond to the y-axis label (ratio of power spectral density (cm^2.km)

LC: I've changed the figure caption to:  Normalized difference of the power spectral density of J2 SLA as a function of wavelength

**Figures general:**

- You tend to use the same color bars for all subplots in your images. You could plot one large colorbar at the bottom with labels that have a bigger font size. The resulting white space could be used to make the plot titles larger (see next comment).

- Your subplot titles include information that is repeated multiple times – e.g. in Fig. 7 all subplots have 'Mission j2, cycles. . .' – could this go in the caption? Make the plot headers larger as they are not legible at 100% size.

LC: I increased the size of the plot headers in most of figures to make them more legible.

**Technical comments:**

LC: All technical comments proposed have been taken into account.

L49: at -> et

L57: coming -> upcoming

L88: proposed -> presented

L108: fit -> fitted

L109-110: grammar

Table 2: use consistent notation (comma or dot)

L296: notice -> note

L375: The altimeter SSH using successively each of the IT corrections tested → The altimeter SSH using IT corrections from each model, respectively, . . . (successively is used in a confusing way more than once in the document – check the other occurrences)

L432: ZHAO model -> the ZHAO model

L433: four models, RAY -> four models RAY,. . .

L450: notice -> note

Figure 9a: RRAY -> RAY

---

## Author Comment (AC3) · 17 Sep 2020

**Response to Christopher Unsworth :**

Dear colleague,

Thank you for your interest in this manuscript and for the comments and suggestions you make.

I reply to your comments and suggestions to change hereafter in blue.

Best regards,

Loren Carrere

Description: Carrere et al., present a timely and desired analysis of a range of methods to correct satellite sea level altimetry data for the effects of internal tides. The effects of internal tides on satellite altimetry data are well known to be aliased due to satellite return periods being greater than the dominant periods of internal tides. Carrere et al., select 7 regions of the Earth to do their analysis – notable for their lack of seasonal stratification and the presence of internal tides.

Comparisons are made from the perspective of variance reduction - as no pure "truth" measurement is possible - the only way to know if better results are made is by comparison. Comparisons are made for "free" models, and models with data assimilation. However, it should be noted that some of the data used in the assimilation is also the data used for the comparison – so it is perhaps unsurprising that these show the least variance. LC: That's why I use 2 different altimeter missions for the validation procedure, and the C2 mission is independent from all models tested.

Main Comments: Two other reviewers have picked up typos and inconsistencies, but both say the manuscript is clear. The majority of our reviewers found the manuscript very confusing. Our comments are summarised below.

1. There is a considerable amount of assumed knowledge in the manuscript. It took us a long time to work out that you were trying to use the models to correct the altimetry data. The abstract is especially hard to understand – this needs to be as clear as possible! Try getting a colleague that doesn't know anything about the subject to read it and see if they can work out the purpose, key results and implications of the paper. LC: abstract has been modified to make it clearer.

2. Although the tables and figures (once font sizes have been made large enough to read) communicate the results well – the method of getting the data in the tables is highly unclear. No equations are presented and no description of how comparisons are made across models is given – as the different models may have different resolutions, time stepping and may have been run for different lengths of time it is hard to know if you have made "like for like" comparisons. A separate table, or additional columns in table 1, would be very useful to understand exactly what you have done. LC: font size of figures has been made larger. The equation of the computation of the STD has been added in the text in section 3.2, and an equation has been also added in section 4.2. I added the information about the resolution of the models used in table 1, then all models are described with more details in section 2. Notice that only HYCOM model is a time-stepping model for which a harmonic analysis has been performed to get the tidal atlas.

3. There is a considerable amount of vague and non-scientific terminology used when describing the results, in the discussion and in the conclusions. Phrases and words such as "performs well", "some",

"weaker", "some problems" are very unscientific ways of describe your results – especially considering the manuscript focuses on validation and numerical comparisons of models. If you have the numbers to quantify these statements you use them at all times. The use of the word "significant" is very misleading. No statistical test is described to give that word a scientific meaning. If you have per formed a significance tests on your data you need to detail that because it would help you communicate your results more clearly and forcefully, if not – you need a different word, or a quantification (e.g. a fraction or percentage). LC: I've replaced some of the non-scientific terminology you've pointed out above and removed the word "significant" when not appropriated.

4. Your objectives seem to be incorrect or are misleading. Objective one is stated as: "The objective of this paper is to present a detailed comparison and a validation assessment of these internal tide models using satellite altimetry". As far as we can tell, this is untrue. No validation of the models themselves is provided. You are comparing the internal tide models, but you cannot validate the models per se- as no "truth" measurement is possible. We found this objective confusing – as we tried to find where you had validated your models. Objective 2: "The analysis focuses on the correction of the satellites' measurements from the coherent internal tide signal for the main tidal constituents, M2, S2, K1 and O1". This objective seems more correct based on the results presented. LC: the paper provides a detailed comparison and also a validation assessment procedure of the internal tide models. We don't use any in situ measurements, but altimeter measurements are "truth" measurement of the ocean variability and using this large dataset is a valuable validation database. The methodology consists in the correction of satellites data using the IT models and then computing the variance differences and some spectral estimations as described in section 4.2. The description of the objectives of the paper has been a bit reformulated in the introduction.

We agree with reviewer #2 that the font in the figures needs to be larger, at least 2x as large. We also suggest using a different colour scheme - one that colour-blind people can use. Colourmaps that do this are common on the internet, some examples are given below:

https://colorbrewer2.org/#type=sequential&scheme=BuGn&n=3

https://www.scientificamerican.com/article/end-of-the-rainbow-new-map-scale-ismore-readableby-people-who-are-color-blind/ LC: the font in the figures has been made larger + the colormap of the figures 1-6 + A1-4 + S1-5 has been changed.

The manuscript does present highly relevant work that the community would like to see, however its communication is poor. As it does not appear that re-working of the models or analysis is needed, we think the study needs minor revisions; but the authors need to take care how they communicate their work.

Minor Comments:

Abstract: 1) One sentence does not make a paragraph (this error is present throughout the manuscript) LC: OK noted

2) "In order to access the targeted ocean signal..." what is the targeted ocean signal? This phrase is not used anywhere else in the manuscript and it is not explained. Start with satellite altimetry- and why it has a source of error due to internal tides, and why correcting it is important (in~2-3 sentences). LC : targeted signal is ocean circulation and mesoscale variability . Sentence has been changed.

3) "several geophysical parameters" – the list of geophysical parameters that exist on Earth is enough to fill a book – what do you mean exactly? LC: I'm talking of the geophysical parameters used to

correct altimeter measurements only, they are listed in section 4.1 which describes the altimeter database.

4) I would also include that this work led to the Zaron model being implemented in the GDR standard (if this is true) LC: yes this is true.

Methods: You need to describe your models more consistently. At present the level of detail given to these models is sporadic. What years are they all run for? At what resolutions? What time-stepping? This kind of information is important as it is relevant for how you compare the results. Description how you compare the data is also needed, equations may well help you communicate this clearly. Are any statistical tests used? How many degrees of freedom are allowed when you calculate the variance? LC: I added the information about the resolution of the models used in table 1, then all models are described with more details in section 2. Notice that only HYCOM model is a time-stepping model for which a harmonic analysis has been performed to get the tidal atlas. Some equations have been added in sections 3.2 and 4.2. We compute the variance on each box of the map and the number of points is different within each box, but we assume a gaussian distribution of the samples in each box.

Discussion: The paragraph startling line 597: "Following the results presented here, a recommendation." This is very interesting, but it unclearly worded. You say the results presented here (as in now) led to the Zaron model being implemented in the GDR standard, which was decided a couple of years ago. This is very significant and important. You could reframe the manuscript as the scientific justification for that decision (which has obviously been presented at conferences prior to submission to the journal – as is usually done). I would include this decision, and the implications of the increase in capability, in the introduction as well. LC: sentence has been reformulated in the Discussion section, and some words have been added in the abstract.

Break these long sentences up. E.g. "The present study indicates that the use of the altimetry database is a valuable tool to validate models of IT surface signature on the global ocean and particularly it complements efficiently the in situ validation processes which are generally more localized in space/time due to the availability of in situ datasets (Dushaw et al. 2017, 1995; Zaron and Ray 2017)."

Turns into: "The present study indicates that the use of the altimetry database is a valuable tool to validate models of IT surface signature on the global ocean. It particularly complements the in-situ validation processes which are generally more localized in space/time due to the availability of in situ datasets (Dushaw et al. 2017, 1995; Zaron and Ray 2017)." LC: done

A general rule to help is: once sentence has one point, or message. If you have two messages, you need two sentences. This shorter and more direct structure especially helps for non-native-English readers. LC: OK

"In addition, many initiatives are now being conducted to try to better understand and model the non-stationary component of the internal tides. Work is progressing on the modelling of the seasonal and interannual internal tides variability: Zhao (2019), Zaron (2019), Ray (personal communication), Ubelmann (personal communication). Within the SWOT Science Team and other projects, several teams also work on 3D simulations using different general circulation models such as HYCOM, MITgcm, NEMO (CMEMS-Mercator-Ocean project in progress), or even a specific spectral approach (S. Barbot et al., in preparation)." We can understand why you want to say this, but none of this is produced in the manuscript you present, so it is not a conclusion you are able to make. You could discuss these ongoing efforts in the light of your new findings, and the implications of the present work has on these efforts. But it is not a conclusion. LC: I keep the discussion about the ongoing efforts listed above as I think it is important to list them in this paper, but I've changed the sentences for clarification.

---

## Author Comment (AC4) · 17 Sep 2020

Dear Editor,

Thank you for your interest in this manuscript and for the comments and suggestions you make. I reply to your comments and suggestions to change hereafter, with LC prefix.

Best regards, Loren Carrere

Minor comments : "Significantly" implies statistics has been done to show this – please reword if you mean large or major. LC: done

Colour figures: The colourmaps used are not 100% friendly for those with impaired

colour vision. I suggest they are remade using a suitable cmocean map instead. At the same time, make sure the text in the figures is readable. LC : I use larger font for the text on the figures where it was not legible. Colourmap has been changed for figures 1-6, A1-4 and S1-5.

Table 2: M2 should have a dot as decimal placeholder, not comma. LC: OK

Figure 7: please use a diverging colour scheme for difference plots, e.g., blue-white-red. LC: I already use a diverging color-map here: blue-gray-red. If using a blue-white-red as suggested we cannot see easily the differences in the amplitudes of the signals, so I prefer to keep the blue-gray-red colormap.

Please mark the regional maps on the large maps and tell us in the caption where they are. LC: the regional maps are localized on Figure 1. If marking the regional maps on figure 7 it makes the figure very confusing because there is too much information and colors/lines, so I preferred not to add the regions on Figure 7.

Sometimes you write Jason-2, sometimes J2. Please stick to one; I prefer the full name spelled out for all the satellites myself since there is a J2 tidal constituent. LC: OK I put J2 in most places. As the paper is not talking about minor tidal constituents I don't think there is any mis-understanding possible in using J2 for Jason-2.